# Microbial biofilms as living photoconductors due to ultrafast electron transfer in cytochrome OmcS nanowires

Jens Neu [1,2,5] ✉, Catharine C. Shipps [1,2,5], Matthew J. Guberman-Pfeffer[1,2], Cong Shen [1,2], Vishok Srikanth [1,2], Jacob A. Spies [3], Nathan D. Kirchhofer [4], Sibel Ebru Yalcin [1,2], Gary W. Brudvig [3], Victor S. Batista [3] & Nikhil S. Malvankar [1,2] ✉

Light-induced microbial electron transfer has potential for efficient production of value-added chemicals, biofuels and biodegradable materials owing to diversified metabolic pathways. However, most microbes lack photoactive proteins and require synthetic photosensitizers that suffer from photocorrosion, photodegradation, cytotoxicity, and generation of photoexcited radicals that are harmful to cells, thus severely limiting the catalytic performance. Therefore, there is a pressing need for biocompatible photoconductive materials for efficient electronic interface between microbes and electrodes. Here we show that living biofilms of *Geobacter sulfurreducens* use nanowires of cytochrome OmcS as intrinsic photoconductors. Photoconductive atomic force microscopy shows up to 100-fold increase in photocurrent in purified individual nanowires. Photocurrents respond rapidly (<100 ms) to the excitation and persist reversibly for hours. Femtosecond transient absorption spectroscopy and quantum dynamics simulations reveal ultrafast (~200 fs) electron transfer between nanowire hemes upon photoexcitation, enhancing carrier density and mobility. Our work reveals a new class of natural photoconductors for whole-cell catalysis.

Living cells have been incorporated with quantum dots and nanostructures for fluorescent labelling and drug delivery for over two decades[1]. However, light-absorbing nanostructures have not been used to drive catalytic reactions inside of cells due to lack of biocompatibility and high cytotoxicity of foreign materials, such as photosensitizers, inside the cell which often limits the operational efficiency[1]. Furthermore, inherent defects in synthetic photosensitizers cause several problems such as photocorrosion, photodegradation and the generation of photoexcited radicals, which results in low stability, irreproducibility and lack of sustainability of biohybrid materials[2].

Some bacteria produce light-absorbing centers but suffer from low electron transfer efficiency and a lack of durability[1]. Natural electron transfer proteins such as azurins, myoglobin and c-type cytochromes do not show photoconductivity[3,4] due to picosecond carrier lifetimes of the heme iron which typically inhibits any charge separation[5]. Covalently linking artificial photosensitizers to these proteins yields low electron transfer rate on the 10 ns timescale or slower, greatly limiting their applications[6]. Furthermore, it is not feasible to use longer excited state lifetimes, such as electron injection from the triplet states due to rapid degradation caused by reactive oxygen species produced in these processes. Therefore, there is an urgent need to develop novel

[1]Department of Molecular Biophysics and Biochemistry, Yale University, New Haven, CT, USA. [2]Microbial Sciences Institute, Yale University, West Haven, CT, USA. [3]Department of Chemistry, Yale University, New Haven, CT, USA. [4]Oxford Instruments Asylum Research, Santa Barbara, CA, USA. [5]These authors contributed equally: Jens Neu, Catharine C. Shipps. ✉e-mail: jens.neu@yale.edu; nikhil.malvankar@yale.edu

biomaterials capable of ultrafast primary electron transfer to achieve efficient charge separation, followed by sequential secondary electron transfer for long-lived charge separation and charge accumulation[6].

To evaluate the use of engineered living materials as living photoconductors, we chose the electroactive soil organism *Geobacter sulfurreducens* because it has evolved the ability to export electrons, derived from metabolism, to extracellular acceptors such as metal oxides and electrodes in a process called extracellular electron transfer (EET)[7,8]. Bacteria establish direct electrical contact to electron acceptors via micrometer-long, polymerized cytochrome nanowires, called OmcS, which eliminates the need for diffusive redox mediators[7,8] (Fig. 1b). Hemes in the OmcS nanowire form a parallel, slipped-stacked pair with each pair perpendicular (T-stacked) to the next pair, forming a continuous chain over the entire micrometer length of the nanowire[7] (Fig. 1d). The minimum edge-to-edge distances is 3.4–4.1 Å between the parallel-stacked hemes and 5.4–6.1 Å between the T-stacked pairs.

Owing to this evolutionarily optimized OmcS nanowire structure with seamless stacking of hemes, *G. sulfurreducens* can transfer electrons over distances of one hundred times their size by forming more than 100 μm-thick highly-conductive nanowire networks in biofilms[9,10], which enables *G. sulfurreducens* cells to generate the highest current density in bioelectrochemical systems[11]. Owing to large electron storage capacity, cytochromes also confer high supercapacitance to biofilms with low self-discharge and reversible charge/discharge[12]. Moreover, a network of purified nanowires can transfer electrons over distances of 10,000-times the size of a cell[9]. Therefore, *G. sulfurreducens* serves as an ideal model system for electrocatalysis, metal corrosion and production of fuels[13,14]. It was previously thought that conductive filaments on the surface of *G. sulfurreducens* are pili[15] and a network of pili confers conductivity to *G. sulfurreducens* biofilms[10,13,16]. However, structural, functional and subcellular localization studies revealed that nanowires on bacterial surface are composed of cytochromes[7,8] whereas pili remain inside the cell during EET and are required for the secretion of cytochrome nanowires to the bacterial surface[17,18].

Nanowires could be widespread and their photophysical properties could be physiologically important because many *Geobacter*-like metal-reducing bacteria form highly conductive biofilms[19,20] and are widely distributed at the surface of earth in shallow sediments that contain abundant sunlight and metal oxides[21–23]. The sediments are capable of transporting electrons over centimeters[24] and can convert incident light into electricity[25]. Illuminating visible light on *G. sulfurreducens* cells has been shown to improve their catalytic performance such as increases in metabolic electron transfer to metal oxides[26] or other semiconducting materials by over 8-fold compared to that observed under dark conditions[21]. Furthermore, light-induced bacterial electron transfer correlated well with the rates of microbial respiration and substrate consumption[26]. However, the underlying molecular and physical mechanism for this increased photocatalytic performance has remained unclear.

In addition to light-induced whole-cell catalysis[21,26], artificially expressing cytochrome OmcS in photosynthetic cyanobacteria, increased catalytic performance in diverse processes such as an increase in photocurrent by 9-fold[27], increase in nitrogen fixation by 13-fold[28], and improved photosynthesis due to 60% increase in biomass[29] compared to the wild-type cyanobacteria. These studies highlight the vital role of OmcS in light-driven biocatalysis. However, intrinsic photophysical properties of OmcS, which could account for these catalytic improvements, have not been investigated.

Out of 111 cytochromes in *G. sulfurreducens*, OmcS is the only nanowire-forming cytochrome essential for EET to Fe(III) oxides abundant in subsurface[14]. Indeed, cytochromes abundant in subsurface during uranium bioremediation function similar to OmcS[30]. OmcS is also important for EET to electrodes during initial stages of biofilm growth[14]. OmcS is also required for interspecies electron transfer in *Geobacter* cocultures to conduct "electric syntrophy"[13,31,32]. This interspecies electron transfer via naturally conductive microbial consortia is important in diverse methanogenic and methane-consuming environments that affect global climate[33–35]. Photosynthetic bacterial species have also been shown to perform electric syntrophy with light-driven conversion of $CO_2$ to value-added chemical commodities[2]. However, the components and pathways responsible for such light-driven biocatalysis have not been identified and potential for photoactivity beyond photosynthetic microorganisms remains largely unknown.

We hypothesized that cytochrome nanowires in the biofilms could be photoactive, enabling efficient electronic interface between microbes and electrodes. Here we show that living biofilms of *Geobacter sulfurreducens* use nanowires of cytochrome OmcS as intrinsic photoconductors. Surprisingly, nanowires show photoconductivity with ultrafast, sub-picosecond heme-to-heme electron transfer which could explain their influence on photocatalytic performance mentioned above. These rates are among the highest for excited-state electron transfer in biology[36].

## Results and Discussion

### Photoconductivity in living biofilms made up of OmcS nanowire network

To determine the role of OmcS nanowires in light-induced electron transfer, we used the genetically engineered *G. sulfurreducens* strain CL-1 because it overexpresses OmcS nanowires (Fig. 1b–d) and forms highly conductive and cohesive biofilms that can be easily transferred to multiple surfaces[37] (Fig. 1a). Upon laser photoexcitation (λ = 408 nm) which is specific to the Soret band of c-type hemes[4], biofilm conductance remained ohmic and increased by 72 ± 21% (Fig. 1e, f). These studies show that living *G. sulfurreducens* biofilms can

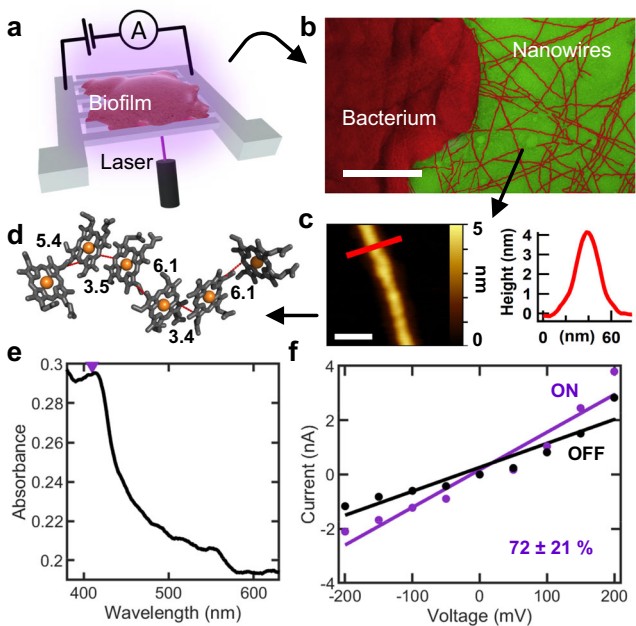

**Fig. 1 | Living photoconductors. a** Measurement schematic. Biofilms are grown on transparent fluorine-doped tin oxide (FTO) electrodes. **b** Transmission electron microscopy of CL-1 cells producing OmcS nanowires. Scale bar, 200 nm. **c** AFM height image of a single OmcS nanowire on mica (left) and respective height profile (right) shown where the red line is indicated. Scale bar 50 nm. **d** Hemes in OmcS stack seamlessly over the entire micrometre-length of nanowires. Edge-to-edge distances are in Å. **e** UV-Visible spectroscopy of biofilm on FTO electrode with the excitation wavelength of 408 nm marked as a purple triangle. **f** Current voltage response of biofilm with the laser on and off. Percentage increase in conductance value represents mean ± standard deviation (S.D.) of two biological replicates. Source data are provided as a Source Data file.

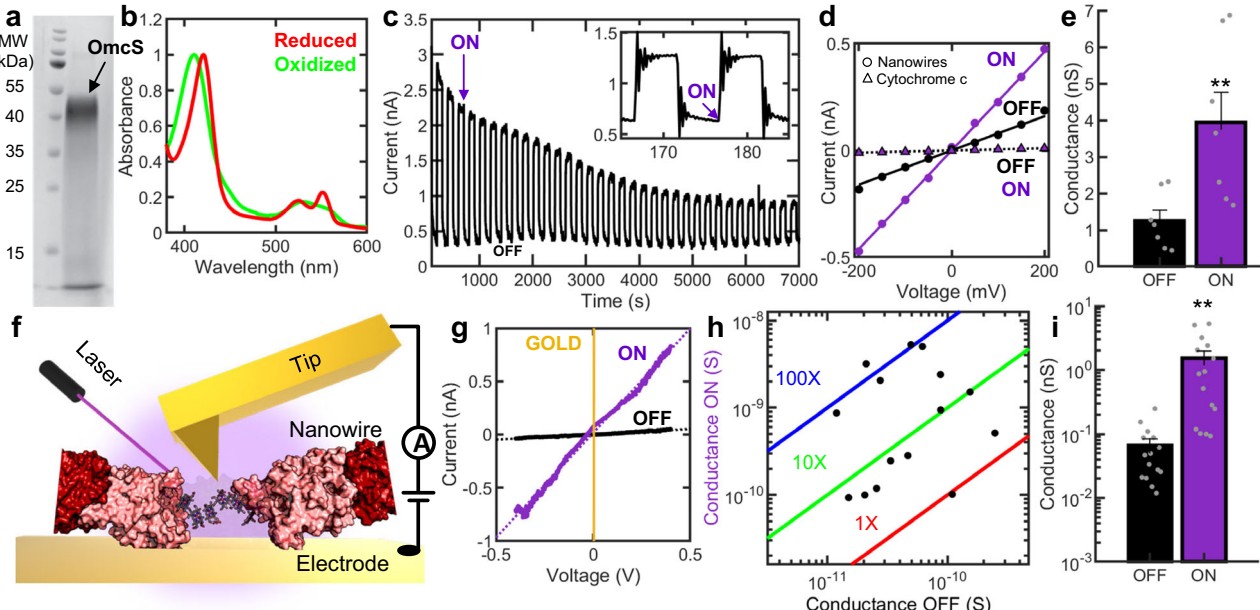

**Fig. 2 | High photoconductivity in purified protein nanowires. a** Heme staining gel of nanowires showing a single band of OmcS. **b** UV-Vis spectrum of oxidized (green) and reduced (red) nanowires. **c** Photocurrent response of nanowire network at 200 mV with the current decay of the off-state subtracted. **Inset:** Fast (<100 ms) photoresponse of nanowires. Axes are same as in Fig. 2c. **d** Current-voltage response of nanowire network and cytochrome c for comparison **e** Comparison of conductance of nanowire network with laser on or off. Values represent mean ± standard error of the mean (S.E.M) with individual data points shown as grey dots (n = 7 independent experiments). ** indicates *p* value = 0.003 using a paired two tail *t*-test. **f** Schematic of pc-AFM of individual nanowires. **g** Current-voltage response of an individual nanowire with a linear fit shown by a purple dashed line. **h** Comparison of conductance increase upon photoexcitation in individual nanowires. Values represent mean of all current-voltage curves measured on individual nanowires (number of curves ranges from 10 to 120 Supplemental Table 2). **i** Comparison of average conductance of individual nanowires with laser on or off. Values represent mean ± S.E.M. with individual data points shown as grey dots (n = 15 independent experiments). ** indicates *p* value = 0.007 using a paired two tail *t*-test. Source data are provided as a Source Data file.

serve as intrinsic photoconductors. As biofilm conductivity determines the bacterial rate of EET[11], our results could explain the increased photocatalytic performance by *G. sulfurreducens*[21,26].

## Rapid photoconductivity in purified nanowires reversible for hours

To determine the origin of photoconductivity in biofilms, we purified nanowires from the CL-1 strain (Fig. 2a). The ultraviolet-visible (UV-Vis) absorbance spectrum of nanowires showed a strong Soret band at 410 nm for air-oxidized nanowires (Fig. 2b). Nanowires were fully oxidized under these conditions because addition of oxidant (ferricyanide) did not change the spectrum (Supplementary Fig. 1a). We placed the nanowires on interdigitated gold electrodes and illuminated from the top (Laser Power = 100 mW/cm²). Photoconductance of nanowire network initially increased more than 6-fold (Fig. 2c), but the extent of conductance increase decreased over time, likely due to laser damage. Nanowires responded faster than 100 ms (Fig. 2c inset). The photoresponse persisted for hours but decreased over time (Fig. 2c). Both the dark current and photocurrent were proportional to an applied voltage ranging from −0.2 to +0.2 V (Fig. 2d), indicating an ohmic conduction behavior of nanowires similar to biofilms. Remarkably, nanowire networks, with and without laser excitation, showed a linear current-voltage response with an average conductance increase of 230 ± 28 % (n = 7), which is higher than common perovskites[38,39] and porphyrin nanowires[40] (Fig. 2d, e).

Multiple control experiments confirmed that the observed photoconductivity is an intrinsic property of nanowires, owing to their polymerized cytochrome architecture. For example, the monomeric horse-heart cytochrome-c showed very low dark current and photocurrent as expected[3,4] when measured under identical conditions (Fig. 2d). Upon addition of a chemical reductant sodium dithionite, the Soret band for reduced nanowires red-shifted to 420 nm as expected[4] (Fig. 2b). These chemically reduced nanowires ($\lambda_{Soret}$ = 420 nm) did

not show significant photoconductance upon excitation at λ = 405 nm, confirming that photoreduction of oxidized hemes are necessary for photoconductivity in nanowires at this excitation (Supplementary Fig. 1b). Switching the electrode material from gold to tungsten also retained photoconductivity, confirming that the measured response is not an artifact of the electrode material (Supplementary Fig. 2). The ratio of laser-on/ laser-off (on/off) current of nanowires increased with increasing laser power, further demonstrating that the measured photoconductivity is solely due to laser excitation (Supplementary Fig. 3). All these experiments together confirm that the nanowires show intrinsic photoconductivity which can account for observed photoconductivity in living biofilms. The difference in photoconductivity between biofilms and purified nanowires is likely due to non-conductive materials such as cells and polysaccharides present in the biofilms.

## Individual nanowires show up to 100-fold photoconductivity increase

To quantify the photoresponse of individual nanowires, we used photoconductive atomic force microscopy (pc-AFM)[41] (λ = 405 nm, Initial Laser Power = 3.20 kW/cm², Fig. 2f). Individual nanowires showed up to 100-fold increase in conductance upon photoexcitation (Fig. 2h–i, Supplementary Table 2). The differences in photoconductance are likely due to variation in the laser power caused due to experimental setup (see methods and Supplementary Fig. 11 for details). The difference in the photoconductivity between individual nanowires and nanowire network is likely due to inter-nanowire as well as nanowire-electrode contact resistance. Notably, the observed 10 to 100-fold increase in conductance for protein nanowires at relatively low bias (< 0.5 V) is substantially greater than that of synthetic porphyrins[42] that show only up to a 5-fold increase at very high bias of 12 V.

These experiments on individual nanowires confirm that the observed photoconductivity response in networks of nanowires is due

to nanowires alone and not because of an artifact of the measurement setup. Furthermore, the observed photoconductivity is not due to heating effects because all pc-AFM experiments were performed in a temperature-controlled environment thus inhibiting any substantial increase in temperature. Furthermore, the linearity and stability of our IV curves indicate that measured conductivity increase is not due to heating (Fig. 2g). In addition, the conductivity of OmcS nanowires decreases upon heating[43] whereas we observed up to 100-fold increase in conductivity upon photoexcitation.

### fs-TA revels sub-picosecond charge separation in nanowire

To understand the mechanism of photoconductivity in protein nanowires, we performed femtosecond transient absorption (fs-TA) spectroscopy by determining the electron dynamics upon photoexcitation on an ultrafast (~100 fs) time scale[5,44] (Fig. 3a). The fs-TA tracks the UV-Vis spectral changes by changing the time delay $\Delta\tau$ between the femtosecond laser pump and the probe pulses and recording a differential absorbance spectrum ($\Delta A$) at each time delay[44] (Fig. 3a). This difference spectrum contains information on the dynamic processes occurring in the system such as excited state energy migration, electron or proton transfer processes and isomerization[44]. In contrast to the above studies of photoexcitation in the Soret band (Figs. 1, 2), we performed fs-TA using excitation in the Q-band ($\lambda = 545$ nm) to avoid thermal damage, and to monitor changes in the region of the strongest absorption bands[44]. It is important to note that Soret and Q-band transitions arise from the same ground state making Q-band excitation a suitable proxy to monitor these processes[44]. Photoexcitation with $\lambda = 530$, 545 and 400 nm yielded similar dynamics, demonstrating a wide spectral range for photoconductivity (Supplementary Figs. 4, 5). Neither buffer alone nor the blank substrate showed any response,

measurements in solid and liquid state are similar, and the ET dynamics were independent of the laser intensity and power (Supplementary Figs. 6, 7, 8) indicating that observed dynamics are due to nanowires and not an artifact of the environment or the substrate.

Upon photoexcitation of protein nanowires, electrons are promoted from the ground state to the excited state, decreasing the ground state population. This decrease caused a negative signal in $\Delta A$ at 410 nm known as a ground state bleach[44] (Fig. 3b, c). In addition, we observed a positive $\Delta A$ which is indicative of excited-state absorption at $\lambda = 367$ nm and $\lambda = 424$ nm after $\Delta\tau = 0.1$ ps and 2 ps, respectively (Fig. 3c, d). These absorptions are absent in the native, air-oxidized, unexcited nanowires (Fig. 3d), indicating that the photoexcitation is causing these absorptions. In particular, the absorption at $\lambda = 424$ nm agrees well with the absorption of chemically reduced nanowires (Fig. 3d), suggesting that upon photoexcitation, excited-state electron transfer is reducing the hemes in the nanowires and thus photoreduction contributes to nanowire photoconductivity.

To understand the origin of different transient oxidation states, we determined the kinetics at the key wavelengths mentioned above using a sequential model[44] that yielded the first excitation timescale of $19 \pm 23$ fs (see methods). This timescale is faster than the instrument response function ($100 \pm 50$ fs) and can, therefore, be treated as an instantaneous excitation on the timescale of the measurement (Fig. 4a). Following this excitation, the charges were transferred between hemes with a decay time of $212 \pm 27$ fs. The corresponding spectra are a superposition of a ground state bleach and the appearance of a new feature around 367 nm, which can be attributed to the doubly oxidized hemes as per the spectral simulations (Fig. 3d). Based on simulations (see below, Fig. 4d), we conclude that the ultrafast charge transfer also results in the formation of a reduced heme in its

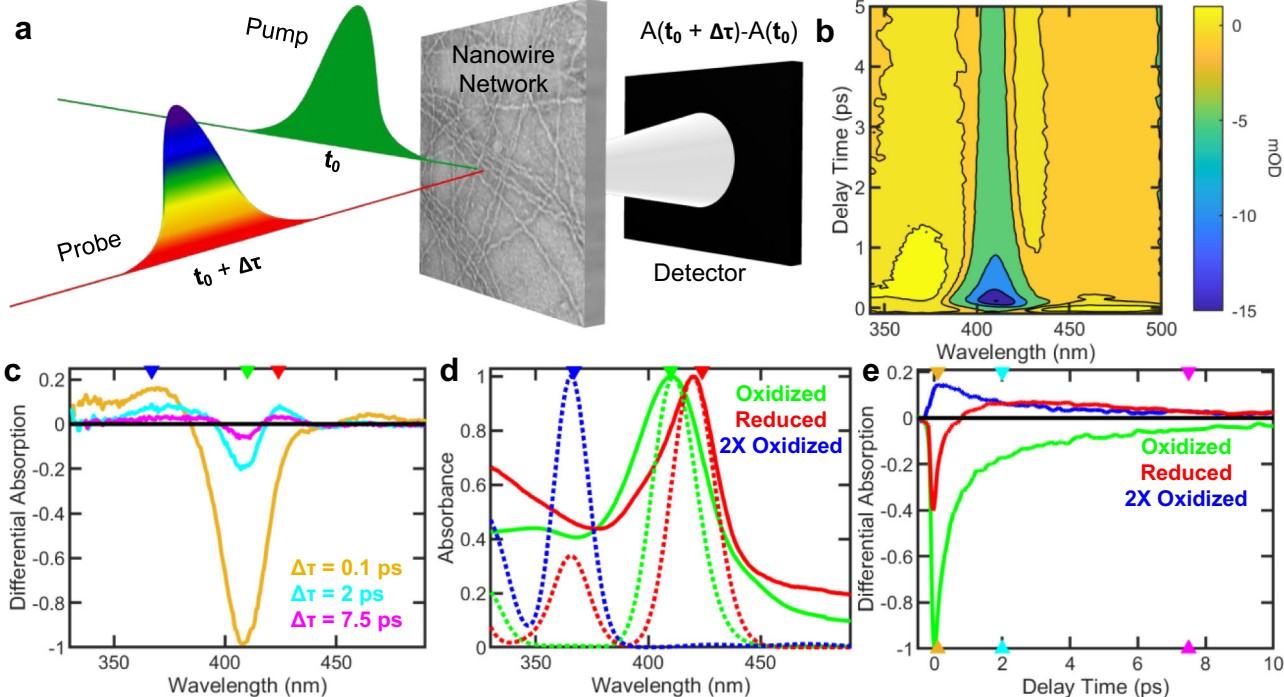

**Fig. 3 | Ultrafast (<100 fs) charge transfer between hemes in nanowires revealed by femtosecond transient absorption spectroscopy (fs-TA). a** Schematic of fs-TA. A pump beam ($\lambda = 545$ nm) excites a nanowire sample and is followed by a probe beam after a time delay. The differential absorption between the initial and time-delayed spectra is detected and reported as optical density. **b** Averaged transient absorption data of nanowires ($n = 6$ independent experiments) where colours represent the milli optical density (mOD). **c** Normalized change in differential absorption with wavelength at different delay times. Key wavelengths are

marked as $\lambda = 410$ nm (green), $\lambda = 424$ nm (red), and $\lambda = 367$ nm (blue). **d** The experimental (solid) and simulated (dashed) spectra of oxidized, reduced, and singlet doubly-oxidized nanowires. Wavelength markers are same as in Fig. 3c. **e** Normalized change in differential absorption over delay time at key wavelengths. Time-markers are shown in the same colour as time traces in Fig. 3c. Traces in **c** and **e** represent mean of $n = 6$ independent experiments. Source data are provided as a Source Data file.

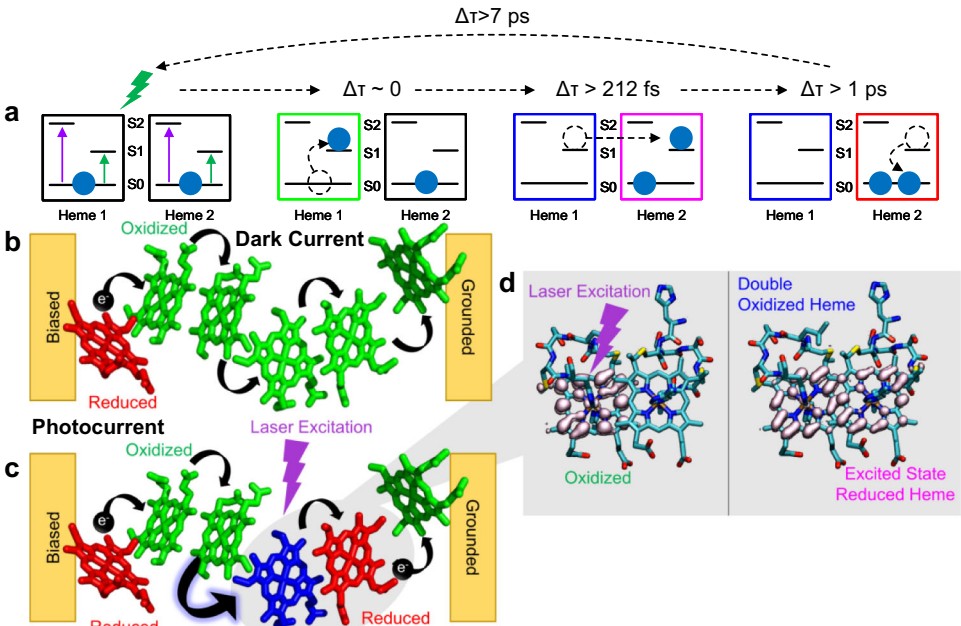

**Fig. 4 | Model for origin of photoconductivity in protein nanowires. a** Simplified energy level diagram for hemes depicting the changes that occur upon photo-excitation in transient absorption and their respective decay times. **b** The dark current in the ground state arises due to propagation of a reduced state created by electron injection from the electrode. **c** The photocurrent is due to the laser excitation initiating an ultrafast charge transfer between hemes, creating newly reduced (red) and double oxidized hemes (blue). The photoreduction provides additional charge carriers and larger driving force for charge transfer, which therefore increases the current under bias. **d** Quantum dynamics simulations of ultrafast charge transfer between hemes in protein nanowires, forming a doubly oxidized heme and an excited state of a reduced heme.

excited state, which is spectroscopically dark. We also found a second decay with a time constant of $1.0 \pm 0.1$ ps that can be attributed to relaxation of the excited reduced heme that increases in absorption at the reduced Soret at $\lambda = 424$ nm. In addition, we found a third decay time of $7.9 \pm 0.3$ ps that can be attributed to the recombination to their initial state, including charge transfers back to the singly oxidized heme ground states.

## Computations suggest excited state electron transfer between hemes

We further compared our experimental UV-Vis spectra of nanowires with time-dependent density functional theory calculations of hemes in the nanowire (Fig. 3d). The maximum of the computed Soret band in the reduced heme ($\lambda = 420$ nm) is shifted by 9.5 nm to the red of the band maximum for the oxidized heme, in good agreement with the 10.5 nm shift observed experimentally (Fig. 3d). These computational analyses further suggest that photoexcitation causes reduction of hemes in the nanowires. Our finding is consistent with prior studies of photoreduction of monomeric cytochromes mediated by the light-induced excited state of hemes even in the absence of external electron donors[45,46].

To evaluate the transient kinetics data obtained using the sequential model fitted to experimental data, we performed quantum dynamics simulations at the Extended Hückel level of theory[47,48]. We simulated the propagation of an electron wavepacket in the excited state from hemes in the nanowires, in a slip-stacked as well as in T-stacked orientation[7]. Our simulations suggest a ~100 fs timescale for photo-induced charge transfer between the slip-stacked pair of hemes (Fig. 4d). This timescale agrees with the experimentally determined timescale for the excited state charge transfer ($212 \pm 27$ fs). The survival probability for electron transfer in a slipped-stack heme pair remains low (<60%) for most energy levels, indicating a high probability for electron transfer to a nearby heme within 100 fs (Supplementary Fig. 9). Thus, the timescale for electron transfer between slipped-stack heme pairs remains similar for most energy levels.

## Hemes are likely electron source for observed photoreduction

As no external electron donor was added, our results suggest that the additional electrons that reduce the heme are intrinsic to the nanowire itself. We further analyzed the possibility that surrounding protein causes the observed photoreduction of OmcS hemes. Several aromatic amino acids, including tryptophan and tyrosine, are within 5 Å of the hemes in the OmcS. Although excitation of either tryptophan or tyrosine is not possible at the wavelengths used in this study[45,46], we considered a possibility that electron transfer can quench a photo-excited heme in a manner similar to flavins in a cryptochrome[49]. This quenching would reduce a heme and leave behind an amino acid radical. The most likely amino acid candidate for radical formation is tryptophan because its radicals have absorbance which would explain the 367 nm species[50]. While the formation of such radicals is possible, the signal strength in fs-TA measurements is determined by the molar extinction coefficients ($\varepsilon$) of the (transient) species. The molar extinction coefficient of the Soret band for OmcS is approximately 100 times larger than those of tryptophan radicals[50,51]. The ground state bleach represents all the photoexcited hemes in the OmcS nanowires and the species corresponding to $\lambda = 367$ nm and 424 nm have differential absorptions of ~20 and 10% of the total magnitude, respectively (Fig. 3e). Therefore, the number of tryptophan radicals created from electron transfer needs to be larger than the number of excited hemes in OmcS if the radical species at $\lambda = 367$ nm arises from tryptophan. Such a possibility seems unlikely because only one radical can be created for every quenched excited heme. Thus, the observed spectra cannot be accounted for by amino acid radicals.

We also evaluated the possibilities of other electron sources causing photoreduction. We found that multi-photon processes are absent in our experiments because the ET dynamics were independent of the laser intensity and power (Supplementary Fig. 8). The magnitude of photocurrent is also linear with increased power (Supplementary Fig. 3).

Redox impurities also did not contribute to the measured spectra because of identical dynamics in solution and in solid-state

(Supplementary Fig. 7). Photodegradation also did not change the electron transfer dynamics, only the magnitude of spectra by <10% over two hours.

We therefore considered an alternative possibility that parallel-stacked hemes can serve as an electron donor and acceptor pair (Fig. 4). We hypothesized that the excited state charge transfer is occurring between two neighboring hemes with only one of the hemes being in the excited state. Such charge transfer would result in the appearance of a reduced heme and leave behind a doubly oxidized heme (Fig. 4). The computed UV-Vis spectrum of a doubly oxidized heme indeed showed an absorption maximum at $\lambda = 365$ nm which agrees with the experimentally observed species at $\lambda = 367$ nm. Our computed spectrum of a doubly oxidized heme thus recaptures the blue shift observed in the transient absorption experiment (Supplementary Fig. 10). The qualitative agreement between the computed and experimental spectra is independent of the spin state of doubly-oxidized species such as the singlet and triplet state.

To identify the nature of doubly oxidized species, we performed an analysis of atomic spin populations. We found that the change in the spin populations occurs only on the ligands and not in the iron center. Therefore, our analysis suggests that doubly oxidized species are $Fe^{3+}$ + porphyrin radical which agrees with the observed spectra at 367 nm. These analyses further suggest that the doubly oxidized species are not $Fe^{4+}$ due to lack of change of spin density on the iron center upon additional oxidation of the heme in the $Fe^{3+}$ state (Supplementary Fig. 10 and Supplementary Table 1).

To further evaluate the thermodynamic feasibility of radical heme species, we used the Rehm-Weller cycle. This analysis requires four energetic terms: (1) energy required to form radical heme species (based on iron-porphyrin systems[52]) (1.7 V), (2) the ground state redox potential of OmcS ($-212$ mV)[51], (3) the photon energy used to excite OmcS nanowires ($\lambda = 545$ nm $= 2.3$ eV), and (4) the vibrational energy difference between the ground and excited states, called the Coulomb stabilization energy associated with the intermediate radical ion pair[53] ($\omega_p$) -60 meV. Therefore, the energetics of this process would be $\Delta G_{et} = [1.7$ eV$-(-0.212$ eV$) + 0.06$ eV$]-2.3$ eV $= -0.4$ eV. Thus, $\Delta G_{et} < 0$ for the formation of the radical heme species, making them energetically feasible. Our analysis is a lower estimate for the net energy available for the formation of the radical species. Therefore, in combination with our simulated analysis, our studies suggest that doubly oxidized species are $Fe^{3+}$ + porphyrin radical and nanowires are photoreduced by ultrafast light-induced heme-to-heme charge transfer.

## Proposed mechanism for ultrafast photoconductivity in OmcS nanowire

Based on above results, we propose the following model for the origin of photoconductivity in OmcS nanowires (Fig. 4). This model is focused on the singlet states and not triplet states because these states are spectroscopically dark, and would be less pronounced due to their lower energies. As these nanowires transport charges through seamless stacking of hemes (Fig. 1c), our prior experiments have shown that they can be treated as redox conductors, with the long-range charge transfer governed by a theoretically-predicted hopping mechanism with negligible carrier loss over micrometers[54]. All hemes in the nanowires are initially oxidized and in their ground state as confirmed by UV-Vis spectroscopy (Fig. 2b). Upon applying a bias, electrons are injected from the electrode into the nanowire, creating a reduced state that travels through the nanowire (Fig. 4b). The photoexcitation triggers an ultrafast charge transfer resulting in an additional reduced state that persists for picosecond timescale, without any applied bias, far away from the electrode (Fig. 4c). This newly formed reduced state will have a mobility similar to the electrode-injected state as they both are present in the same nanowire with identical structure. Therefore, upon photoexcitation, the density of reduced states is increased, thus increasing the carrier density of the OmcS to generate

photoconductivity in nanowires. The photoreduction observed in our fs-TA is consistent with this model.

In addition to the higher carrier density due to photogenerated electrons, it is likely that the mobility of electrons increases upon photoexcitation due to increased driving force for charge transfer in the excited state of hemes[43]. Upon photoexcitation an electron is promoted from the ground state to an excited state. The ultrafast charge transfer between neighboring hemes creates a reduced-state heme in the excited state and a doubly oxidized heme (Fig. 4c, d). The reduced-state heme can then relax from the excited to the ground state. Upon photoexcitation, the uniformly oxidized nanowire is thus partially reduced and partially double oxidized (Fig. 4c).

The generated doubly oxidized heme will alter the redox energies of the heme chain, with a more positive redox potential. We have previously found that the redox potential of OmcS hemes becomes substantially positive upon oxidation[43]. The OmcS nanowires transport charges via a hopping mechanism[54]—a process in which a charge (electron or hole) temporarily resides at a heme, changing its redox state. The driving force for charge transfer depends on the redox energies of the electron donating and accepting hemes. Therefore, the charge transfer rate is directly related to the mobility.

For the fully oxidized (non-excited) state, this process initiates at the electrode surface where injected electrons hop to nanowire redox sites, creating locally reduced hemes. For the photoexcited state, this process is enhanced because transferring an electron to the double-oxidized species, and removing an electron from a reduced heme, are significantly more favorable in the illuminated nanowire than for the oxidized nanowire in the dark. The increased likelihood for charge transfer upon photoexcitation will then result in increased mobility. Furthermore, the initial ultrafast charge transfer between hemes increases the lifetime of the photogenerated state. Both the generation of a "new" mobile charge and the increase in its mobility will contribute to the observed increase in conductivity upon photoexcitation.

In summary, we demonstrate, for the first-time, significant photoconductivity in a living system due to ultrafast light-induced charge transfer within protein nanowires. The surprising origin of photoconductivity in these natural systems lies in the higher carrier density and mobility upon photoexcitation.

Although ultrafast electron transfer can occur in monomeric cytochromes, it typically requires incorporated dyes as photosensitizers and sacrificial electron donors[36] which can be toxic to cells[1]. In contrast, we find that the protein nanowires intrinsically exhibit robust and ultrafast charge transfer without any need for such site-selective labeling. Our studies thus establish OmcS nanowires as photoconductors intrinsic to cells with capability of ultrafast electron transfer, thus eliminating the need for foreign materials such as molecular dyes or inorganic nanoparticles that limit the catalytic performance[1].

Furthermore, our studies show that sub-ps charge transfer is possible in natural proteins in an excited state. Prior ultrafast electron transfer studies have reported the ground state rates of 15–90 ps in the closest-stacked hemes[36]. This difference is likely because excited-state rates are known to be faster due to higher energy and larger orbital delocalization compared to the ground-state rates[49].

Although many bacterial EET studies remain focused on electrons, protons play a very important role, not only in bacterial energy generation, but also in the electronic conductivity of proteins[55]. For example, through measurements of the intrinsic electron transfer rate, we previously found that both the energetics of a glutamine (proton acceptor) and its proximity to a neighboring tyrosine (proton donor), regulate the hole transport over micrometers in amyloids through a proton rocking mechanism[56]. Therefore, it is very important to couple electron/proton transfer to accelerate EET and for the development of electronically conductive protein-based biomaterials.

The high surface area of these nanowires, combined with their biocompatibility and lack of toxicity, make them attractive candidates for an emerging field of light-driven whole-cell bioelectrocatalysis for a wide range of applications such as water splitting, chemical sensing and $CO_2$ fixation and production of chemicals, fuels and materials[57]. Our studies may also help establish the efficient and stable production of liquid fuels from sunlight using a liquid sunlight approach[5]. Future studies on nanowires with different heme stacking and protein environment[8] or substituting the metals from iron to zinc[58] or tin[59] could vary the interactions between the heme cofactors to alter the electronic and photophysical properties of nanowires for tuneable functionality[57].

## Methods

### Bacterial biofilm growth and OmcS nanowire purification

*Geobacter sulfurreducens*[60] strain CL-1, which produces an elevated abundance of OmcS protein[37], was obtained from our laboratory culture collection and grown on electrodes in a bioelectrochemical system as previously described[20,61]. For the growth in liquid culture, cells were grown until stationary phase[61] and collected via centrifugation, and then a slightly modified version of a previously described protocol[7] was used to shear extracellular filaments from the cells. In brief, pelleted cells were suspended in 150 mM ethanolamine pH 10.5 and blended for 2 min on low speed in a commercial unit (Waring). Cells and cell debris were removed by centrifugation, first at 13,000 and then at 23,000x *g*. OmcS filaments were then collected either by precipitation in 12.5% ammonium sulfate or ultracentrifugation at 100,000x *g*, in accordance with previously described protocols for obtaining microbial nanowires from *G. sulfurreducens*[10]. Collected OmcS filament samples were resuspended and stored in 150 mM ethanolamine pH 10.5 and dialyzed to remove residual ammonium sulfate where appropriate.

### UV-Vis Spectroscopy

UV-Vis spectra were recorded with a spectrophotometer (Avantes AvaSpec-ULS2048CL-EVO). For nanowires, a quartz slide was cleaned with ethanol and 2 μl of 80 μM protein was dropped on this slide and then dried for 20 min in the desiccator. Another 2 μl was dropped on the same spot and again dried in the desiccator for 20 min. The spectrum was collected for the air oxidized sample. Then 40 mg/ml of sodium dithionite in water was dropped to cover the protein spot (2–3 μl). The dithionite caused a chemical reduction of the hemes in the protein. The spectrum of the reduced material was then recorded. All spectra were normalized such that the minimum and maximum absorbance values for wavelengths above 380 nm were set to zero and 1, respectively. The biofilm solid state measurements were taken on a FTO electrode in hydrating conditions with the background of a clean FTO electrode subtracted.

### Electrode fabrication

Three different types of electrodes, based on gold (Au), Tungsten (W) and fluorine doped tin oxide (FTO), were used. The designs consisted of interdigitated electrodes, which create a "finger" pattern in which every odd-numbered line is connected to one pad, and every even numbered to the opposite electrical contact. This dense electrode packing ensures a large number of electronic contacts. The measured data are averaging over 132 wire connection pairs and provides superior signal compared to a single electrode device.

For the gold-electrodes, the spacing between each line was 5 μm, and for the tungsten and FTO-electrodes, the spacing was 10 μm. For all cases, the electrode (the metalized part) was 10 μm wide.

The gold and tungsten electrodes were fabricated using UV-Lithography on thermally oxidized Silicon wafer. The thermal oxidation created a 300 nm silicon oxide layer which provides a plain and electrically insulating substrate. The metal electrodes were fabricated by spin coating a double resist, consisting of LOR 5-A and S1805. LOR 5-A was coated at 3000 rpm for 1 min followed by 5 min heating at 180 °C. Following this baking step, a second resist layer S1805 was applied at 3000 rpm for 1 min and cured at 120 °C for 2 min. Afterwards, the resists were exposed to UV radiation through a shadowmask and developed in MIF 319 developer for 2 minutes. The structured photoresists were then metalized using 5 nm Ti or Cr and 40–60 nm Au or W. A lift-off in heated (80–120 °C) NMP removed the metalized resists and resulted in the final microstructure electrode. A protection coating was then spin coated onto the device. This coating was washed off with acetone before using the electrode. Each electrode was tested prior to protein deposition to ensure proper electric insulation between the two electrodes.

For the FTO IDE electrodes, commercially available FTO on Quartz glass was used. S1805 resist was spin coated and structured as previously described. After structuring this resist was used as a soft mask in reactive ion etching. The etching was carried out in an Oxford Plasmalab 100 RIE with a chamber pressure of 8 mTorr, and a gas flow of 8 sccm $Cl_2$ and 40 sccm Ar. The etching was carried out until the unwanted FTO was completely removed. The remaining photoresist was cleaned off in hot NMP (120 °C) and the final devices was covered with protective coating. This coating was removed with acetone prior to usage of the electrodes. Each electrode was carefully checked to ensure that the two contacts are electrical insulated.

### DC Conductance of Biofilms and Nanowires

Conductivity measurements on nanowires and biofilms were performed as described previously[62]. Connections to device electrodes were made with a probe station (MPI TS50) inside a Dark Box which formed a Faraday cage and also blocked background light. Current and voltage were applied using a semiconductor parameter analyzer with preamplifiers (Keithley 4200 A-SCS) allowing for 1 fA current and 0.5 μV voltage resolution. Two-point DC conductance measurements utilized two probe needles to contact the device on two adjacent electrodes. A fixed voltage in the range of ±0.3 V was applied to the two electrodes for a minimum of 100 s in sampling mode until a steady current was reached. Voltage-current points were fit with a line, and the slope was used to determine conductance (G).

Devices were prepared by dropping 0.5 μL of 8 μM nanowires in 150 mM ethanolamine pH 10.5 on to the device and let dry in ambient atmosphere overnight. The drop formed on the material had a diameter of 1.4 ± 0.1 mm. The electrode area was 2 × 2 mm, which ensured that all the material was electrically contacted.

To perform photoconductivity measurements, the previously described probe station was equipped with a diode laser, with an average output flux of 100 mW/cm$^2$ and a center wavelength of 408 nm. This laser spot was adjusted to be larger than the electrode area which ensured a homogenous excitation of the material. The laser beam was blocked/released using an optical shutter with a 1 ms response time.

Conductance measurements on reduced nanowires was performed by mixing 0.25 μL of a concentrated solution of sodium dithionite with 9.75 μL of nanowires in an anaerobic environment such that there was 50-fold molar excess of dithionite to heme concentration in the final solution. 0.5 μL was dropped onto an electrode and dried in the anaerobic chamber overnight.

### Transient Absorption (TA) Data Collection

The transient absorption spectra were collected at the Center for Functional Nanomaterials (CFN), part of the Brookhaven National Laboratory. Further data was collected at Drexel University. The signal to noise ratio of the commercial CFN spectrometer was superior to the data from Drexel, hence the data collected at Drexel were solely used in the supplement of this manuscript. The detailed description refers to the data collection at CFN.

The TA spectra were collected using a Helios (Ultrafast Systems) TA spectrometer. The excitation wavelength was generated in a TOPAS OPA. The probe pulse was generated via supercontinuum in calcium fluoride.

For each measurement, spatial overlap was optimized for strongest signal. Each set of data was iterated for several hours. Each iteration was then compared to the mean of the iteration to ensure long-time stability of the spectrometer and the sample material.

The sample was prepared by drop casting 5 µl of protein solution onto a freshly cleaned quartz substrate. The samples were allowed to dry for 60 minutes in a desiccator. This deposition was repeated to create thicker films. Based on the optical transmission a location on the sample with sufficient Soret band absorption and acceptable scattering was chosen.

### Transient Absorption (TA) Data Processing

The collected TA spectra were processed using three softwares: Surface Xplorer (Ultrafast Systems), MATLAB, and Glotaran. Surface Xplorer was used to visualize the data and select measurements with adequate signal to noise. This selection reduced the number of processed spectra to 15. Of these measurements seven were pumped at 545 nm (shown in the main text) four pumped at 530 nm (shown in SI) and four pumped at 400 nm (shown in SI). All these measurements were evaluated to determine kinetics and dynamics of the spectral evolution. The main evaluation was performed for a pump wavelength of 545 nm, as this was depositing the lowest energy and therefore heating into the sample material. The other two wavelength confirmed the determined kinetics.

Surface Xplorer was used to compensate for spectral chirp associate with wavelength dependent dispersion in the used sample and the quartz substrate. This correction ensured that the time zero point was independent of the wavelength. After this initial processing, the measured 1024 wavelength points were adjacent averaged to 512 points, resulting in a wavelength resolution of approximately 1 nm.

The preprocessed data were then imported into the MATLAB. Six measurements at 545 nm pump were averaged into a single set (after accounting for time-zero jitter). These data sets are shown in the main text. The dynamics at 410 nm, 367 nm, and 424 nm was simultaneously fitted with a double exponential function convoluted with the instrument responds function and an instantaneous injection model as Heaviside function. The lifetimes from this simple three wavelength dynamic fit are used as starting points for the detailed target analysis using Glotaran.

The preprocessed data (from the Surface Xplorer data processing) were then loaded into Glotaran and truncated to −2 ps to infinity in time and 340-505 nm in the wavelength space. This software was used for global analysis. The model assumes a global decay dynamic defined by a fixed number of decay constants. Based on our model we decided that a sequential analysis is best suited to describe the processes in photoexcited OmcS.

The sequential model, however, cannot directly separate the individual species from the ground state bleach of the main species. This is caused by the temporal overlap between the decays and the parallel decay into the ground state from the excited heme species which is not completely undergoing a charge separation step.

The sequential model assumes an excitation, which fitted to $19 \pm 23$ fs. This is faster than the instrument responds function of $100 \pm 10$ fs. The excitation can therefore be considered to be instantaneous (justifying the used Heaviside approximation in the preliminary analysis in MATLAB). Following the excitation, the charges are transferred from one heme to another within $212 \pm 27$ fs. The corresponding spectra are a superposition of a ground state bleach and the appearance of a new feature around 367 nm, which is identified as a double

oxidized heme (according to the presented spectral simulation). This charge transfer results in the formation of a reduced heme in the excited state. A second decay with a time constant of $1 \pm 0.1$ ps describes the relaxation of the excited reduced heme. A final third time constant with $7.9 \pm 0.3$ ps describes the relaxation of the system to its initial state, including charge back transfer to the single oxidized heme ground state.

### Photoconductive AFM (pc-AFM)

Topography and electrical conductivity of nanowires on the gold surface were both measured using conventional tapping (AC) and conductive atomic force microscopy (c-AFM, ORCA™) measurement modes with a commercially available AFM (Cypher ES, Oxford Instruments Asylum Research, USA) equipped with blueDrive™ photothermal excitation. The probe was a commercially available ASYELEC-01-R2 probe (Asylum Research) with Ti/Ir coating and nominal resonant frequency $f = 75$ kHz, spring constant $k = 2.8$ N/m, and tip radius $R_{tip} = 28 \pm 10$ nm; measured values were $f_O = 86.6$ kHz and $k_O = 5.8$ N/m for the specific probe used in these measurements. In order to bias the sample, a small neodymium magnet (1/32" x 1/16" diam., K&J Magnetics) was both adhered and electrically contacted to the top surface of the nanowires-on-gold sample using silver paint (PELCO® Leitsilber, Ted Pella).

For tapping mode topography, the probe was driven with piezo actuation at a scan rate of 1 line/s, a free amplitude of 120 nm (0.58 V at a sensitivity of 207 nm/V), and a -100 nm setpoint (0.5 V) to keep the tip-sample interaction very gentle in the so-called "attractive" or "non-contact" state to avoid damaging the nanowires.

Following the topographical scan, cAFM was used with a force setpoint of 50 nN to execute point I-V measurements on individual nanowires to measure their conductivity ($n = 15$ nanowires), with a sample voltage sweep of ±0.5 V at 1 V/s sweep rate for 20 sweep cycles at 2 kHz acquisition rate (1 kHz lowpass filter). The additional effects of photoexcitation on nanowire conductivity were examined by toggling the blueDrive™ laser as the excitation source (405 nm, 10 mW DC, with a spot diameter of $2 \pm 1$ µm) on at least 20 sequential I-V sweeps. This was accomplished using the provided 0.01X filter cube (Asylum Research) and positioning the laser spot at the very apex of the probe tip (rather than using it as the probe oscillatory excitation); this provides $[10e\text{-}3\,W]/[\pi*(1e\text{-}6\,m)^2]*0.01 = 32$ µW/µm² illumination on the nanowires.

For control experiments in these pc-AFM measurements, a fresh template stripped gold sample (identical to the surface that the nanowires were deposited on) was also prepared with an electrical contact as above. As a positive control, the tip-gold conductivity was measured at the same conditions (50 nN loading force, ±0.5 V at 1 V/s, 20 cycles) to ensure an ohmic contact in the absence of nanowires. As a negative control for photoconductive measurements, the tip-gold conductivity was measured at the same conditions (50 nN loading force, ±0.5 V at 1 V/s, 20 cycles) with the blueDrive excitation sequentially toggled on and off to confirm that there was no change in tip-gold conductivity from the 405 nm excitation in the absence of nanowires.

**pc-AFM data analysis.** At each collection point on an individual nanowire, at least 20 IV curves were collected. The last half of all the current voltage curves collected (minimum 10 curves) at a single point were used to calculate conductance. The IV curves were then sorted by nanowire and the slope of each curve was measured to get the conductance. For all nanowires, any outliers in conductance were removed by three median absolution deviation analysis on the $\log_{10}$ of conductance. All the remaining individual conductance values for each nanowire were averaged to get the mean conductance of the single nanowire. The analysis for the laser OFF and laser ON current voltage curves was identical.

## Laser characterization

When interpreting photoexcitation experiments it is crucial to verify whether the experiment was performed in a linear or non-linear excitation regime. In the later the photoexcitation would be strong enough to trigger non-linear effects (i.e., saturable absorption) or cause electron-electron interaction (e-e scattering, Auger-effect, etc.). These effects would make a conclusive discussion of the experiments more challenging. In the linear regime, only a small percentage of molecules is excited, while the majority remains in their ground state. While there is no ultimate threshold for the linear vs non-linear regime, it is common to accept less than 1% excitation as linear.

We calculated the percentage excitations based on the known optical power density 100 mW/cm², and the total lifetime of the photoexcited system ($\tau = 7.9$ ps). The core concept used here is that for any given time point a certain number of photons hit the hemes and excite them while previously excited hemes are recombining into their ground state.

The recombination is described as $N(t) = N(t-\Delta t) e^{-\Delta t/\tau}$, with $\Delta t$ as small time step[63]. The CW laser excitation was discretized using the time step to yield a total photon flux set by the laser power. Assuming a 100% quantum yield, this means that the generation of excited hemes is described directly by the photon flux. Starting at $t = 0$, the population rises in competition with the recombination as $N(t) = G(\Delta t) - N(t-\Delta t) e^{-\Delta t/\tau}$. After a time of a nanosecond, $N(t)$ approaches a quasi-steady state value of $1.6 \times 10^6$ molecules/cm². Comparing this value to the total protein density of $2.5 \times 10^{13}$ molecules/cm² yields a ratio $6.4 \times 10^{-6}$ %. This approximated value is well below 1%, justifying the linear interpretation of our experiments.

## Computational methods

**OmcS structure modeling.** The *c*-type heme cofactors of OmcS were modelled as iron porphyrin, with the methyl, thioether, and propionic acid substituents of the macrocycle replaced by hydrogen atoms. The two axially coordinated histidine residues were truncated at the $C_b–C_g$ bond to give 1-methylimidazole ligands. This model system has been extensively used to theoretically characterize the structures, spectra, and reactivity of heme cofactors[64].

The geometry of the heme model was optimized at the density functional theory (DFT) level in the reduced, singly oxidized, and doubly oxidized redox states. Harmonic frequency analyses confirmed that the heme model was at a local minimum on the respective ground state potential energy surface for each redox state. The reduced and singly oxidized species were optimized with the lowest spin multiplicity (singlet and doublet, respectively). The doubly oxidized heme model was examined in both the triplet and singlet manifolds. For the ground-state singly and doubly oxidized species, the expectation value of the spin-squared operator, $<S^2>$, was 0.75, and 2.00 after annulation of spin contaminates.

All geometry optimizations and harmonic frequency analyses were performed with the Becke, three-parameter, Lee–Yang–Parr (B3LYP) hybrid functional[65] and a mixed basis set, applying the LANL2DZ effective core and valance functions to Fe[66], and the 6-31 G(d) basis to H, C, and N atoms. As with the vertical excitation calculations described in the next subsection, we employed tight self-consistent field convergence and an ultrafine integration grid, as implemented in Gaussian 16 revision A.03.

The simulation was performed on two types of adjacent heme pairs, namely T-Stack and slip stack present in the OmcS structure (Fig. 1d). Only the slip-stack pairs exhibited charge transfer (Supplementary Fig. 9b). Therefore, this calculation was limited on neighboring hemes.

**Simulated absorption spectra.** The absorption spectrum of the heme in each redox state was simulated *in vacuo* with time dependent (TD)-DFT using the B3LYP functional and a 6-31 + G(d) basis set for all atoms[67–69]. The predicted spectra were uniformly shifted by 38 nm to improve the alignment with the experimental spectra. The excited states of interest—the two Soret transitions—exhibited some spin contamination for the singly and doubly oxidized species, which is a well-known issue with TD-DFT[70]. However, $<S^2>$ deviated by only 0.2–0.5 from the uncontaminated value, and the blue-shift predicted for the doubly oxidized species relative to the reduced or singly oxidized species was similar irrespective of whether it was modeled as a closed-shell singlet or open-shell triplet. We therefore conclude that the blue shift needed to explain the experimental observation is independent of the spin contamination present in our open-shell calculations.

**Quantum dynamics simulations.** The dynamics of photoinduced intermolecular electron transfer between adjacent hemes was modeled with a previously described wavepacket propagation methodology implemented within the tight-binding Extended Hückel framework[47]. This level of theory was previously used to describe the electronic structure of iron-, as well as other, metalloporphyrins[48].

Structures used for both optical spectra simulations and quantum dynamics simulations are provided as Supplementary Data 1.

## Statistics & reproducibility

Sample sizes were based upon accepted conventions within the field to ensure reproducibility and statistics and no explicit power analysis were conducted. No data were excluded from analysis. All experiments were independently repeated multiple times defined in the legend and all attempts to replicate the experiments were successful. Randomization was not relevant to the study as all samples were treated similarly either for nanoscale or bulk studies. Investigators were not blinded to group allocation during data collection or analysis as all samples were treated similarly either for nanoscale or bulk studies. Images shown in Fig. 1b, c were repeated at least three times. Gel shown in Fig. 2a was repeated at least three times.

## Reporting summary

Further information on research design is available in the Nature Research Reporting Summary linked to this article.

## Data availability

The datasets generated during and/or analysed during the current study are available from the corresponding author on reasonable request. The key relevant datasets generated during and/or analysed during the current study are included along with the paper. All other relevant data are included in the Supplementary Information. Source data are provided with this paper.

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

## Acknowledgements

We thank Derek Lovley for providing the strain. We also thank Jason Baxter (Drexel University) for making his TA spectrometer available for preliminary measurements. This research was supported by a Career Award at the Scientific Interfaces from Burroughs Welcome Fund (to N.S.M.), the National Institutes of Health Director's New Innovator award (1DP2AI138259-01 to N.S.M.), and NSF CAREER award no. 1749662 as well as EAGER award no. 2038000 (to N.S.M.). Research was sponsored by the Defence Advanced Research Project Agency (DARPA) Army Research Office (ARO) and was accomplished under Cooperative Agreement Number W911NF-18-2-0100 (with N.S.M). This research was also supported by an All Points West grant and a NDSEG Graduate Research Fellowship award (to C.C.S.). This research used resources of the Center for Functional Nanomaterials (CFN), which is a U.S. Department of Energy Office of Science User Facility, at Brookhaven National Laboratory under Contract No. DE-SC0012704, as well as the Yale SEAS cleanroom, the Yale West Campus cleanroom, and the Yale West Campus Imaging Core.

## Author contributions

J.N. and N.S.M designed bulk experiments whereas N.K., S.E.Y. and N.S.M. designed nanoscale experiments. J. N. fabricated electrodes and performed fs-TA with J.A.S., J.N. and C.C.S. measured conductivity of nanowires and biofilms C.C.S. measured UV-Vis spectra, M.J.G. performed computation under the supervision of V.S.B, C.S. grew biofilms on electrodes in microbial fuel cell and fabricated electrodes, V.S. purified protein nanowires, N.K. and S.E.Y. performed pc-AFM, G.W.B helped with data interpretation and N.S.M. supervised the project. J.N. C.C.S. and N.S.M. wrote the manuscript with input from all authors.

## Competing interests

The authors declare no competing interests.
