## [Peer Review File · Nature Communications]

Microbial biofilms as living photoconductors due to ultrafast electron transfer in cytochrome OmcS nanowiresREVIEWER COMMENTS

Reviewer #1 (Remarks to the Author):

The authors present evidence of photoconductivity in living biofilms of *Geobacter sulfurreducens* and consider possible sources of photoconductivity using experiments and theory. Photoconductivity is probed using current-voltage response measurements on biofilms and nanowire networks and photoconductive AFM measurements on individual nanowires. The biofilm and biofilm components showed an increase in conductivity upon photoexcitation. To look into possible sources of this photoconductivity, the authors performed fs-TA on the nanowire networks. These spectra together with TD-DFT analysis indicate the appearance of reduced and double oxidized hemes upon photoexcitation of the oxidized nanowires. The authors simulated the propagation of an excited state electron wave packets across the hemes and found a high probability of electron transfer. They compared their simulated charge transfer timescales with experiment and found agreement. The authors suggest the photoconductivity is due to an increase in the carrier density due to photoreduction of the nanowire and an increase in mobility of the electrons upon photoexcitation. The writing is confusing in places, some figure descriptions are unclear, and the SI does not provide hoped for details in many cases. The paper might be appropriate for publication if the specific concerns below are addressed, and the overall clarity of the presentation is improved:

- 1) On p. 1 the authors say “Natural electron transfer proteins such as c-type cytochromes do not show photoconductivity due to fast (\sim ps) charge recombination of the heme iron”. This is not correct. Hemes have rapid non-radiative decay channels and are not deactivated by “charge recombination” per se.
- 2) Is there any evidence that *Geobacter* uses photoconductivity properties in its natural function?
- 3) P. 6. Please elaborate on the photoreduction mechanism. What molecular species are likely being photoexcited and what is the nature of the possible electron transfer process(es) at play? Is this heme-to-heme charge transfer? Are other species involved? Formation of doubly oxidized hemes seems very unlikely. Please discuss this hypothesis in the context of the well-known redox potentials of heme proteins and the energy scales associated with the excitation.
- 4) It also seems very unlikely (p.7) that coherent transport underpins charge transport in the heterogeneous environment of the protein. Indeed, Blumberger and many others have simulated transport in multi-heme proteins (see recent paper of J Butt et al. in PNAS 2021) and found that the transport is well described by incoherent multi-step hopping, as would be expected. I am skeptical of

the proposed coherent mechanism. Hopping is indeed mentioned on p. 8. Is the coherent model actually used, or is a hopping model employed. This is not described well in the body of the paper or in the SI.

5) The linkage between photoconductivity and photocatalysis mentioned in the abstract should be fleshed out or deleted. I do not think that photoconductivity properties are needed to produce valuable photocatalysts, so the photoconductivity that is being explored – while significant – is being oversold in the context of catalysis.

6) Fig. 2h lists $n=20$, but I think $n=15$ based on the SI and counting the data points; I think it would be helpful to add more information on this figure to the manuscript. I found Fig. 3e confusing with regards to the time-markers.

7) Please comment on differences in on/off ratios seen between biofilms, nanowire networks, and individual nanowires; and do the authors have thoughts on the variability of the individual nanowires, where some show 10-fold while others show up to 100-fold increases in photoconductivity?

8) I suggest the authors included more information on their kinetics model, either in the manuscript or SI. What was the scale of this model? It looks like it was two hemes based on the SI, but this is not described explicitly. Do the authors expect larger models to show significant differences? Did they try other dynamics simulations? What was the reasoning for including the slip-stacked and T-stacked orientations? Please explain.

9) The description of the increase in electron mobility at the end of the manuscript should be improved (around line 170)

10) The authors mention metal substitution as a way of tuning functionality, which is an interesting suggestion, but there is no mention of using other nanowires with different structures. Do the authors anticipate that nanowires with different structures may also show high photoconductivity and could this be another avenue for tuning living photoconductors?

11) At the start of the manuscript the authors mentioned that “Photoconductive atomic force microscopy shows >10 fold current on/off ratio...”. Please elaborate on this “on/off” ratio. Later in the manuscript, this term has frequently, so it should be described at the outset.

12) In Fig. 4 (a), a simplified energy diagram is shown to describe the electron transfer mechanism, where S₀, S₁, and S₂ states are involved. Is there any possibility for triplet state involvement? Can the author comment on this?

13) Regarding the mechanistic aspects of protein nanowire photoconductivity, what role might the protein environment play? Also, is there a possible role for proton transfer processes in this mechanism?

Reviewer #2 (Remarks to the Author):

Referee report on “Living photoconductors based on ultrafast electron transfer in microbial nanowires” by Jens Neu et al., submitted to Nature Communications.

Electroactive organisms such as the model system *Geobacter sulfurreducens* are attracting growing attention in diverse research communities due to their particular electrical properties and for their future potential application in emerging domains such as biological electrical materials and bioelectronics.

The eye-catching highlight of this manuscript is that the authors demonstrate photoconductivity in living biofilms of *Geobacter sulfurreducens*, which is not only a novelty from a fundamental point of view, but in the long term could also open the door to optoelectronic applications.

The claimed photoconductivity and the ultrafast electron transfer is obtained from a variety of state-of-the-art measurement techniques ranging from macroscopic interdigitated test structures to photoconductive atomic force microscopy and femtosecond transient absorption spectroscopy. Besides the experimental results obtained with this measurement methodology, the authors also introduce quantum dynamics simulations and propose a model for the origin of photoconductivity in protein nanowires, suggesting that upon photoexcitation, excited-state electron transfer is reducing the hemes in the nanowires. An exemplary powerful result of this latter approach is that both the experimentally and computed Soret bands in the reduced heme are shifted by around 10 nm to the red of the band maximum for the oxidized heme. The manuscript is overall well written and also the quality and exceptional didactics of the figures are highly appreciated.

The authors are invited to address the following questions and suggestions :

Already in the abstract the authors claim that they show REVERSIBLE photoconductivity in living biofilms of *Geobacter sulfurreducens*. The photoresponse shows indeed an on and off level in respectively the presence and absence of light. However the behaviour is not completely reversible, since from figure 2.C the photocurrent response of nanowire network at 200 mV shows a clear decrease as a function of time. On line 56 the authors state that the photoresponse persisted for hours but decreased over time. However, no further discussion or interpretation is given to this decreasing behaviour. The authors are invited to elaborate somewhat more on this decrease of photoresponse over time and on the not completely reversible nature of the photoconductivity. Is this for instance related to ageing/degradation or is this an intrinsic effect ?

By illuminating samples and substrates the temperature of both could rise. How can the authors be sure that the observed effects are only due to illumination and not to induced thermal effects ? How are temperature effects ruled out ? Did the authors measure the local temperature of samples/substrates under illumination ? For instance, a power of 3.20 kW/cm² as used for pc-AFM is gigantic compared to solar illumination (1 kW/m²).

On page 8 of the manuscript the authors state/pose that upon photoexcitation the carrier density increases and “it is likely that the mobility of electrons increases”. Also here the authors are invited to address/discuss the influence of temperature.

On line 88 the authors state that they performed fs-TA using excitation in the Q-band ($\lambda = 545$ nm) to avoid thermal damage etc. This is in contrast to the earlier presented results where photoexcitation in the Soret band was studied ($\lambda = 405$ nm), motivated by the statement that the Soret and Q-band transitions arise from the same ground state (line 90). Did the authors perform the same experiments as in figures 1 and 2 under 545 nm illumination to be sure that indeed the proposed approach and interpretations/results are valid ?

The authors state on line 98 that they observed “a positive ΔA which is indicative of excited-state absorption at $\lambda = 367$ nm and $\lambda = 424$ nm”, which “are absent in the native, air-oxidized nanowires (Fig.2b)”. A small note here is that the x-axis of Fig.2b starts close to 400 nm and therefore the signal at 367 nm cannot be seen in the figure in the present form and therefore cannot support the previous statement.

Based on the high level of novelty and excellent quality of the manuscript, we recommend the editors of Nature Communications to accept this manuscript for publication with minor revisions.

Responses to reviewer comments. Comments are in bold and changes to manuscript are in italics.

Reviewer #1 (Remarks to the Author):

The authors present evidence of photoconductivity in living biofilms of *Geobacter sulfurreducens* and consider possible sources of photoconductivity using experiments and theory. Photoconductivity is probed using current-voltage response measurements on biofilms and nanowire networks and photoconductive AFM measurements on individual nanowires. The biofilm and biofilm components showed an increase in conductivity upon photoexcitation. To look into possible sources of this photoconductivity, the authors performed fs-TA on the nanowire networks. These spectra together with TD-DFT analysis indicate the appearance of reduced and double oxidized hemes upon photoexcitation of the oxidized nanowires. The authors simulated the propagation of an excited state electron wave packets across the hemes and found a high probability of electron transfer. They compared their simulated charge transfer timescales with experiment and found agreement. The authors suggest the photoconductivity is due to an increase in the carrier density due to photoreduction of the nanowire and an increase in mobility of the electrons upon photoexcitation.

The writing is confusing in places, some figure descriptions are unclear, and the SI does not provide hoped for details in many cases. The paper might be appropriate for publication if the specific concerns below are addressed, and the overall clarity of the presentation is improved:

We apologize for the confusion. We have now rewritten the manuscript and figure descriptions to clarify the points of confusion. More details have been added in the manuscript and SI and all specific concerns raised by the reviewer have been addressed below and in the revised manuscript.

1) On p. 1 the authors say “Natural electron transfer proteins such as c-type cytochromes do not show photoconductivity due to fast (~ ps) charge recombination of the heme iron”. This is not correct. Hemes have rapid non-radiative decay channels and are not deactivated by “charge recombination” per se.

We apologize for the confusion due to use of our term “charge recombination”. As suggested by the reviewer, we have replaced this term by “carrier lifetime” as follows:

Natural electron transfer proteins such as c-type cytochromes do not show photoconductivity due to short (~ ps) carrier lifetimes of the heme iron which inhibits any charge separation.

2) Is there any evidence that *Geobacter* uses photoconductivity properties in its natural function?

As suggested by the reviewer, we have now added detailed discussion about how *Geobacter* uses photoconductive properties in its natural function (Ref. 20-28) as follows:

*Many *Geobacter*-like metal-reducing bacteria form highly conductive biofilms^{1,2} and are widely distributed at the surface of earth in shallow sediments which contain abundant sunlight and metal oxides³⁻⁵. The sediments are capable of transporting electrons over centimeters⁶ and can convert incident light into electricity⁷. Illuminating visible light on *G. sulfurreducens* cells have been shown to improve its catalytic performance such as increase in metabolic electron transfer to metal oxides⁸ or other semiconductive materials by over 8-fold compared to that observed under dark conditions³. Furthermore, light-induced bacterial electron transfer correlated well with the rates of microbial respiration and substrate consumption⁸. However, underlying molecular mechanism for this increased performance has remained unclear.*

In addition to light-induced whole-cell catalysis^{3,8}, artificially expressing cytochrome OmcS in photosynthetic cyanobacteria, increased catalytic performance in diversity of processes such as an increase in photocurrent by 9-fold⁹, increased nitrogen fixation by 13-fold¹⁰ and improved photosynthesis

by increasing 60 % biomass¹¹ compared to the wild-type cyanobacteria. These studies highlight the important role of OmcS in light-driven biocatalysis. However, intrinsic photophysical properties of OmcS, which could account for these catalytic improvements, have not been investigated.

The OmcS is also required for interspecies electron transfer in *Geobacter* cocultures to carry out “electric syntrophy”¹²⁻¹⁴. This interspecies electron transfer via naturally conductive microbial consortia is important in diverse environments¹⁵⁻¹⁷. Photosynthetic bacterial species have also been shown to perform electric syntrophy with light-driven conversion of CO₂ to value-added chemical commodities¹⁸. However, the components and pathways responsible for such light-driven biocatalysis have not been identified and potential for photoactivity beyond photosynthetic microorganisms remains largely unknown.

3) P. 6. Please elaborate on the photoreduction mechanism. What molecular species are likely being photoexcited and what is the nature of the possible electron transfer process(es) at play? Is this heme-to-heme charge transfer? Are other species involved?

We have clarified in the manuscript that our excitation ($\lambda=410$ nm) is *specific* to the hemes and our experiments and computations show that the measured conductivity is due to heme-to-heme charge transfer. We do not find any evidence of species other than hemes participating in charge transfer.

Formation of doubly oxidized hemes seems very unlikely. Please discuss this hypothesis in the context of the well-known redox potentials of heme proteins and the energy scales associated with the excitation.

We have further clarified in the manuscript that our transient absorption spectroscopy studies reveal a novel feature at 367 nm which agrees with our computational results of doubly oxidized heme. As laser-driven changes are highly non-equilibrium, it is difficult to relate them directly with known redox potentials which are thermodynamic equilibrium values.

4) It also seems very unlikely (p.7) that coherent transport underpins charge transport in the heterogeneous environment of the protein. Indeed, Blumberger and many others have simulated transport in multi-heme proteins (see recent paper of J Butt et al. in PNAS 2021) and found that the transport is well described by incoherent multi-step hopping, as would be expected. I am skeptical of the proposed coherent mechanism. Hopping is indeed mentioned on p. 8. Is the coherent model actually used, or is a hopping model employed. This is not described well in the body of the paper or in the SI.

We apologize for this confusion. We have now performed additional studies and revised the text to clarify that we have not used coherent model. The OmcS contains a heme chain comprised of alternating T-stack and slip stack hemes (Fig. 1b). The ultrafast charge transfer was observed only in the slip stack pair and not in the T-stack pair (Supplementary Fig. 7b). As charge transport through OmcS nanowires requires charge transfer between all hemes, it will be unlikely a fully coherent charge transport. In the revised manuscript, we have also discussed the studies mentioned by the reviewer.

5) The linkage between photoconductivity and photocatalysis mentioned in the abstract should be fleshed out or deleted. I do not think that photoconductivity properties are needed to produce valuable photocatalysts, so the photoconductivity that is being explored – while significant – is being oversold in the context of catalysis.

We apologize for the confusion. As suggested by the reviewer, we have rewritten the abstract by not mentioning photocatalysis to avoid confusion.

In the revised manuscript, we have clarified that although photoconductivity is not essential for photocatalysis in many systems, conductive nanowires enable *Geobacter* to transport electrons over

hundreds of cell lengths and generate highest current density in biocatalytic systems. Our studies show that the photoconductivity of OmcS nanowires could account for many previous photocatalysis studies using *Geobacter* as a catalyst or artificially expressing OmcS in photosynthetic microorganisms to improve biocatalytic processes such as increase in photocurrent to minerals or other materials as well as improved nitrogen fixation and photosynthesis (Ref. 20-28).

Fig. 2h lists n=20, but I think n=15 based on the SI and counting the data points; I think it would be helpful to add more information on this figure to the manuscript. I found Fig. 3e confusing with regards to the time-markers.

We apologize for this error and have now corrected the legend. Each of the 15 points shown in 2h is an individual nanowire that we performed at least 20 different current voltage sweeps from -1 to +1 V. Of these data collected, the last half (at minimum 10) were used to calculate the average conductance of each nanowire (See Methods in Supplement). We have corrected the legend to say $10 \leq n \leq 120$ and added a table to the supplement to reflect this (Supplementary Table 1).

We thank the reviewer for their input regarding Figure 3e. We wanted to illustrate how the time cuts and the spectral cuts are related to each other. The arrows in e refer to the time cuts in c, and vice versa. We now added the arrows on top and bottom to clarify points of confusion.

7) Please comment on differences in on/off ratios seen between biofilms, nanowire networks, and individual nanowires;

We have clarified in the revised manuscript that *the difference in photoconductivity between biofilms and purified nanowires is likely due to non-conductive materials such as cells and polysaccharides present in the biofilms*. These materials absorb and scatter the incoming light, but do not contribute to photoconductivity.

We have also clarified in the revised manuscript that *the difference in the photoconductivity between individual nanowires and nanowire network is likely due to inter-nanowire as well as nanowire-electrode contact resistance*.

and do the authors have thoughts on the variability of the individual nanowires, where some show 10-fold while others show up to 100-fold increases in photoconductivity?

We have added a new figure (Supplementary Fig. 8) and have clarified in the revised manuscript that the variation in the photoconductivity of individual nanowires is *likely due to variation in the laser power caused due to limitation of experimental setup* which we have now explained in detail in our methods. The initial power of the laser in AFM is 3.20 kW/cm² which is the minimum power available in this setup. As AFM measurements take long time of several minutes, this setup minimizes sample damage by avoiding direct illumination of nanowires as the laser is scattered off the cantilever. (Supplementary Fig. S8). However, this scattering causes some variation in the laser power that can cause the variation in photoconductivity that could account for variations observed in the photoconductivity (Fig. 2).

8) I suggest the authors included more information on their kinetics model, either in the manuscript or SI. What was the scale of this model? It looks like it was two hemes based on the SI, but this is not described explicitly. Do the authors expect larger models to show significant differences? Did they try other dynamics simulations? What was the reasoning for including the slip-stacked and T-stacked orientations? Please explain.

As suggested, we have included more information about the kinetic model in the methods. We have now clarified that both slip-stacked, and T-stacked heme pairs were included to mimic the OmcS structure (Fig. 1d). We do not expect that a use of larger models will change the result of the kinetic modeling because ultrafast charge transfer is limited only to the slipped stacked pair, owing to its smaller edge-to-

edge distance (~ 3.4 Å) than T-stacked heme pairs (~ 5.4 Å) (Supplementary Fig. 7b). Due to large size of OmcS protein, it is not feasible to use other models such as non-adiabatic quantum mechanical/molecular mechanical dynamical simulations.

The simulation was performed on two types of adjacent heme pairs, namely T-Stack and slip stack present in the OmcS structure (Fig. 1d). As shown in figure S7b, only the slip-stack pairs exhibited charge transfer, which is the reason why this calculation was limited on neighboring hemes.

9) The description of the increase in electron mobility at the end of the manuscript should be improved (around line 170)

As suggested, we have described our recent theoretical and experimental studies on the mobility of OmcS nanowires (Ref. 41) and have added more discussion about the increase in electron mobility as follows: In addition to the higher carrier density due to photogenerated electrons, it is likely that the mobility of electrons increases upon photoexcitation *due to increased driving force for charge transfer by altered redox state of hemes. The driving force for charge transfer depends on the redox energies, which depend on the redox states of the electron donating/accepting heme. Therefore, the charge transfer rate is directly related to the mobility.*

10) The authors mention metal substitution as a way of tuning functionality, which is an interesting suggestion, but there is no mention of using other nanowires with different structures. Do the authors anticipate that nanowires with different structures may also show high photoconductivity and could this be another avenue for tuning living photoconductors?

As suggested we have now added discussion about nanowires with different heme stacking arrangement (Ref. 9). Our studies show that the distance between hemes is key to ultrafast electron transfer (Fig. 4). Therefore, we expect that nanowires with different structures can show photoconductivity depending on their heme geometry.

11) At the start of the manuscript the authors mentioned that “Photoconductive atomic force microscopy shows >10 fold current on/off ratio...”. Please elaborate on this “on/off” ratio. Later in the manuscript, this term has frequently, so it should be described at the outset.

We have clarified this term at the beginning of the manuscript, by explicitly mentioning “laser induced” and “laser-on/laser-off (on/off)”, for the first two occasions that term is used. We have clarified legend entries that use on/off to refer to measurements of both “On” and “Off” states by explicitly stating they were performed with the laser on or off.

12) In Fig. 4 (a), a simplified energy diagram is shown to describe the electron transfer mechanism, where S0, S1, and S2 states are involved. Is there any possibility for triplet state involvement? Can the author comment on this?

As suggested, we have now added the discussion about triplet states as follows:

This model is focused on the singlet states and not triplet states because these states are spectroscopically dark, and would be less pronounced due to their lower energies.

13) Regarding the mechanistic aspects of protein nanowire photoconductivity, what role might the protein environment play?

We have discussed previous work that shows the effect of protein environment on the stacking of hemes (Ref. 9). For example, lowering the pH brings heme closer that could increase the electron transfer rate.

Also, is there a possible role for proton transfer processes in this mechanism?

We have now included a discussion about the role of proton transfer in conductivity as follows:

Although many bacterial EET studies remain focused on electrons, protons play a very important role, not only in bacterial energy generation, but also in the electronic conductivity of proteins²⁰. For example, through measurements of the intrinsic electron transfer rate, we previously found that both the energetics of a glutamine (proton acceptor) and its proximity to a neighboring tyrosine (proton donor), regulate the hole transport over micrometers in amyloids through a proton rocking mechanism²¹. Therefore, it is very important to couple electron/proton transfer to accelerate EET and for the development of electronically conductive protein-based biomaterials.

Reviewer #2 (Remarks to the Author):

Referee report on “Living photoconductors based on ultrafast electron transfer in microbial nanowires” by Jens Neu et al., submitted to Nature Communications.

Electroactive organisms such as the model system *Geobacter sulfurreducens* are attracting growing attention in diverse research communities due to their particular electrical properties and for their future potential application in emerging domains such as biological electrical materials and bioelectronics.

The eye-catching highlight of this manuscript is that the authors demonstrate photoconductivity in living biofilms of *Geobacter sulfurreducens*, which is not only a novelty from a fundamental point of view, but in the long term could also open the door to optoelectronic applications.

The claimed photoconductivity and the ultrafast electron transfer is obtained from a variety of state-of-the-art measurement techniques ranging from macroscopic interdigitated test structures to photoconductive atomic force microscopy and femtosecond transient absorption spectroscopy. Besides the experimental results obtained with this measurement methodology, the authors also introduce quantum dynamics simulations and propose a model for the origin of photoconductivity in protein nanowires, suggesting that upon photoexcitation, excited-state electron transfer is reducing the hemes in the nanowires. An exemplary powerful result of this latter approach is that both the experimentally and computed Soret bands in the reduced heme are shifted by around 10 nm to the red of the band maximum for the oxidized heme. The manuscript is overall well written and also the quality and exceptional didactics of the figures are highly appreciated.

The authors are invited to address the following questions and suggestions:

Already in the abstract the authors claim that they show REVERSIBLE photoconductivity in living biofilms of *Geobacter sulfurreducens*. The photoresponse shows indeed an on and off level in respectively the presence and absence of light. However the behaviour is not completely reversible, since from figure 2.C the photocurrent response of nanowire network at 200 mV shows a clear decrease as a function of time. On line 56 the authors state that the photoresponse persisted for hours but decreased over time. However, no further discussion or interpretation is given to this decreasing behaviour. The authors are invited to elaborate somewhat more on this decrease of photoresponse over time and on the not completely reversible nature of the photoconductivity. Is this for instance related to ageing/degradation or is this an intrinsic effect ?

As suggested, we have discussed this decreasing response over time in the revised manuscript.

Photoconductance of nanowire network initially increased more than 6-fold (Fig. 2c), *but the extent of conductance increase decreased over time, likely due laser damage.*

By illuminating samples and substrates the temperature of both could rise. How can the authors be sure that the observed effects are only due to illumination and not to induced thermal effects ? How are temperature effects ruled out? Did the authors measure the local temperature of samples/substrates under illumination ? For instance, a power of 3.20 kW/cm² as used for pc-AFM is gigantic compared to solar illumination (1 kW/m²).

We apologize for the confusion. In the revised manuscript, we have clarified that only a small fraction of the initial laser power is used for photoexcitation (Supplementary Fig. 8).

We have also revised the manuscript to clarify that the observed photoconductivity is solely due to illumination as followed

The ratio of laser-on/ laser-off (on/off) current of nanowires increased with increasing laser power, further demonstrating that the measured photoconductivity is solely due to laser excitation (Supplementary Fig. 3).

We have also added following text to clarify that the observed photoconductivity is not due to induced thermal effects.

Furthermore, the observed photoconductivity is not due to heating effects because All pc-AFM experiments were performed in a temperature-controlled environment thus inhibiting any large increase in temperature. Furthermore, the linearity and stability of our IV curves indicate that measured conductivity increase is not due to heating (Fig. 2g). In addition, the conductivity of OmcS nanowires decreases upon heating³² whereas we observed up to 100-fold increase in conductivity upon photoexcitation

On page 8 of the manuscript the authors state/pose that upon photoexcitation the carrier density increases and “it is likely that the mobility of electrons increases”. Also here the authors are invited to address/discuss the influence of temperature.

As suggested, in addition to comments by Reviewer 1 (Comment #9), we have rewritten this section with additional discussion as outlined in our response to the Reviewer 1.

On line 88 the authors state that they performed fs-TA using excitation in the Q-band ($\lambda = 545$ nm) to avoid thermal damage etc. This is in contrast to the earlier presented results where photoexcitation in the Soret band was studied ($\lambda = 405$ nm), motivated by the statement that the Soret and Q-band transitions arise from the same ground state (line 90). Did the authors perform the same experiments as in figures 1 and 2 under 545 nm illumination to be sure that indeed the proposed approach and interpretations/results are valid?

The smaller excitation cross-section at 545nm results in a weaker photoconductivity response which precludes us from measuring photoconductivity using Q-band excitation. Therefore, we instead performed fs-TA at both 400 nm and 545 nm to confirm that the same electron transfer dynamics occur at both excitations (Supplementary Fig. 6).

The authors state on line 98 that they observed “a positive ΔA which is indicative of excited-state absorption at $\lambda = 367$ nm and $\lambda = 424$ nm”, which “are absent in the native, air-oxidized nanowires (Fig.2b)”. A small note here is that the x-axis of Fig.2b starts close to 400 nm and therefore the signal at 367 nm cannot be seen in the figure in the present form and therefore cannot support the previous statement.

We apologize for this error. Figure 3d shows a larger range of the nanowire spectra, covering the mentioned wavelengths. We changed the reference to Fig. 3d.

Based on the high level of novelty and excellent quality of the manuscript, we recommend the editors of Nature Communications to accept this manuscript for publication with minor revisions.

REVIEWER COMMENTS

Reviewer #1 (Remarks to the Author):

There are many claims and assertions in this paper that I find problematic, so I cannot support publication.

The authors invoke cytochrome P450 chemistry (refs 46,47) to justify the generation of high oxidation state species ($\text{Fe}(+4)$) in the cytochromes. This seems unreasonable since the multi-heme cytochromes at play are not P450 like in their structure or function. The authors also suggest a role for coherent delocalization of charge across multiple hemes (ref 50), which seems unlikely given the well understood and weak inter-heme couplings in these proteins, and the structural disorder that is present. As well, the authors' claims seem to be inconsistent with finding of Blumberger and others for these multi-heme proteins. Subpicosecond heme-to-heme electron transfer claimed in the manuscript is at odds with recent studies of van Wonderen et al. (PNAS 118, e2107939118) that find nanosecond time scales for activationless heme-to-heme electron transfer. My suspicion is that the photochemistry observed may involve multi-photon processes or redox active impurities in the samples, rather than direct heme-to-heme photoinduced charge transfer with unusual redox states (i.e., $\text{Fe}(+4)$). Perhaps the processes at play arise from the photodegradation pathways cited in the manuscript. The events being studied, while interesting, may not originate in simple heme-to-heme charge flow at all. Typical nonradiative decay rates of heme singlet excited state are sub-nanosecond, so simple inter-heme electron transfer from heme excited states seems unlikely.

Reviewer #2 (Remarks to the Author):

Referee report on revised manuscript "Microbial biofilms as living photoconductors due to ultrafast femtosecond heme-to-heme electron transfer in cytochrome nanowires" by Jens Neu et al., submitted to Nature Communications.

The initial title of the manuscript was "Living photoconductors based on ultrafast electron transfer in microbial nanowires". From my perspective, the eye-catching highlight of this initial manuscript was "that the authors demonstrated photoconductivity in living biofilms of *Geobacter sulfurreducens*, which is not only a novelty from a fundamental point of view, but in the long term could also open

the door to optoelectronic applications. The claimed photoconductivity and the ultrafast electron transfer is obtained from a variety of state-of-the-art measurement techniques ranging from macroscopic interdigitated test structures to photoconductive atomic force microscopy and femtosecond transient absorption spectroscopy. “

Although an extensive – and still rapidly growing - amount of reports can be found on the electrical transport properties and electrical transport mechanisms in *Geobacter Sulfurreducens* and in other electro-active microorganisms, the study of the photoconductive properties in only starting – see next paragraph - and therefore highly novel.

Only a limited number of papers on the subject are found in literature, including the following :

For *Geobacter Sulfurreducens*, Zhang et al. (ref. 20) demonstrated in 2021 that visible-light illumination could alter the electronic state of OM c-cyts from the ground state to the excited state in vivo.

Van Wonderen, et al. (2019 – ref. 34) studied Ultrafast Light-Driven Electron Transfer in a Ru(II)tris(bipyridine)-^[11]_{SEP}Labeled Multiheme Cytochrome using^[11]_{SEP} ultrafast transient absorbance spectroscopy, to define heme-heme electron transfer dynamics in the representative multiheme cytochrome STC (tetra-heme Cytc) from *Shewanella oneidensis* in aqueous solution. STC was photosensitized by site-selective labeling with a Ru(II)(bipyridine)dye and the dynamics of light-driven electron transfer described by a kinetic model corroborated by molecular dynamics simulation and density functional theory calculations.

Furthermore, in 2021 Van Wonderen et al. reported transient absorbance spectroscopy when a His/Met-ligated heme was introduced at a defined site within the decaheme extracellular MtrC protein of *Shewanella Oneidensis*, observing rates of heme-to-heme electron transfer on the order of $10^{*}9s^{-1}$ (in good agreement with their predictions based on density functional and molecular dynamics calculations). This latter paper is not mentioned in the current manuscript of Jens Neu et al. and we'll come back to this paper in the next paragraphs.

The current manuscript of Jens Neu et al., uses transient absorption spectroscopy (as van Wonderen et al.), but furthermore introduces photoconductive AFM (pc-AFM) in the study of photoconductivity of *Geobacter Sulfurreducens*. To the best of my knowledge, this is the very first report using pc-AFM on electro-active microorganisms. This innovative combination of pc-AFM and the other mentioned techniques to explore the still unknown area of photoconductivity in *Geobacter Sulfurreducens*, as reported in the first part of the manuscript, is from my perspective the most appreciated part of the paper due to its high pioneering research level.

In the second part of the paper, the authors introduce TD-DFT calculations, quantum dynamics simulations, and propose a model for the origin of photoconductivity in protein nanowires,

suggesting that upon photoexcitation, excited-state electron transfer is reducing the hemes in the nanowires with a “ultrafast femtosecond heme-to-heme electron transfer” as extra emphasized in the new title of the paper. (This “ultrafast femtosecond heme-to-heme electron transfer” in the new title is somewhat misleading as the used instrument response time is 100 +/- 15 fs.) Furthermore, as also indicated by the other referee, this time-scale differs significantly from the “Nanosecond heme-to-heme electron transfer rates in multiheme cytochrome nanowire..” reported by van Wonderen et al. (PNAS2021-Vol.118-N°39-e2107939118), already mentioned above. An important drawback of the current version of the manuscript in relation to the proposed interpretation of the origin of the observed photoconductivity, is that a thorough discussion with these earlier results is missing, and that no reference is made to this paper of van Wonderen.

We also agree with the other referee that to invoke cytochrome P450 chemistry (ref 46-47) to justify the generation of high oxidation state species (Fe(+4)) in the cytochromes under study in this paper, is doubtful due to structural and functional differences, and therefore possibly other processes are in play.

The authors also state in the conclusions that “the surprising origin of the photoconductivity in these natural systems lies in the higher carrier density and mobility upon photoexcitation”. However the provided argumentation for an increased mobility is not clearly elaborated. Also the hypothesis that “such ultrafast electron transfer might explain the quantum effects recently predicted in these OmcS nanowires (ref.50)” lacks argumentation.

Although the first part of the paper was highly appreciated for its innovative experimental approach, the second part of the revised paper – although ambitious and with good intentions - does not provide at the moment sufficient convincing and strong arguments to support the formulated conclusions regarding the heme-heme electron transfer.

In conclusion, I would expect that a (major or new) revised version of the manuscript addressing the objections formulated above, could lead to a version suitable for publication in Nature Communications.

Point-by-point response to the reviewers' comments. Comments are in bold and changes to the manuscript are in a smaller font here and highlighted in yellow in the revised manuscript.

Reviewer #1 (Remarks to the Author):

There are many claims and assertions in this paper that I find problematic, so I cannot support publication. The authors invoke cytochrome P450 chemistry (refs 46,47) to justify the generation of high oxidation state species (Fe(+4)) in the cytochromes. This seems unreasonable since the multi-heme cytochromes at play are not P450 like in their structure or function.

We apologize for this confusion. Although we did not directly claim that double oxidized species are Fe^{4+} , we recognize the confusion created by our citations that also show similarity to our spectra. We have removed these citations to avoid confusion. We have also performed additional analysis that suggests that the high oxidation species are not Fe^{4+} , rather Fe^{3+} + porphyrin (Supplementary Fig. 10 and Table 1). We have rewritten the discussion about potential mechanisms for the observed photoconductivity as follows:

As no external electron donor was added, our results suggest that the additional electrons that reduce the heme are intrinsic to the nanowire itself. We further analyzed the possibility that surrounding protein causes the observed photoreduction of OmcS hemes. Several aromatic amino acids, including tryptophan and tyrosine, are within 5 Å of the hemes in the OmcS. Although excitation of either tryptophan or tyrosine is not possible at the wavelengths used in this study^{43,44}, we considered a possibility that electron transfer can quench a photoexcited heme in a manner similar to flavins in a cryptochrome⁴⁷. This quenching would reduce a heme and leave behind an amino acid radical. The most likely amino acid candidate for radical formation is tryptophan because its radicals have absorbance which would explain the 367 nm species⁴⁸. While the formation of such radicals is possible, the signal strength in fs-TA measurements is determined by the molar extinction coefficients (ϵ) of the (transient) species. The molar extinction coefficient of OmcS's Soret band is approximately 100 times larger than those of tryptophan radicals^{48,49}. The ground state bleach represents all the photoexcited hemes in the OmcS nanowires and the species corresponding to $\lambda=367$ nm and 424 nm have differential absorptions of ~20% and 10% of the total magnitude, respectively. Therefore, the number of tryptophan radicals created from electron transfer needs to be larger than the number of excited hemes in OmcS if the radical species at $\lambda=367$ nm arises from tryptophan. Such a possibility seems unlikely because only one radical can be created for every quenched excited heme. Thus, the observed spectra cannot be accounted for by amino acid radicals.

We also evaluated the possibilities of other electron sources causing photoreduction. We found that multi-photon processes are absent in our experiments because the ET dynamics were independent of the laser intensity and power (Supplementary Figure 8). The magnitude of photocurrent is also linear with increased power (Supplementary Figure 3).

Redox impurities also did not contribute to the measured spectra because of identical dynamics in solution and in solid-state (Supplementary Figure 7). Photodegradation also did not change the electron transfer dynamics, only the magnitude of spectra by < 10% over two hours.

We therefore considered an alternative possibility that parallel-stacked hemes can serve as an electron donor and acceptor pair (Fig. 4). We hypothesized that if the excited state charge transfer is occurring between two neighboring hemes with only one of the hemes being in the excited state. Such charge transfer would result in the appearance of a reduced heme and leave behind a doubly oxidized heme (Fig. 4). The computed UV-Vis spectrum of a doubly oxidized heme indeed showed an absorption maximum at $\lambda=365$ nm which agrees with the experimentally observed species at $\lambda=367$ nm. Our computed spectrum of a doubly oxidized heme thus recaptures the blue shift observed in the transient absorption experiment (Supplementary Figure 10). The qualitative agreement between the computed and experimental spectra is independent of the spin state of doubly-oxidized species such as the singlet and triplet state.

To identify the nature of doubly oxidized species, we performed an analysis of atomic spin populations. We found that the change in the spin populations occurs only on the ligands and not in the iron center. Therefore, our analysis suggests that doubly oxidized species are Fe^{3+} + porphyrin radical which agrees with the observed spectra at 367 nm. These analyses further suggest that the doubly oxidized species are not Fe^{4+} due to lack of change of spin density on the iron center upon additional oxidation of the heme in the Fe^3 state (Supplementary Figure 10 and Table 1).

To further evaluate the thermodynamic feasibility of radical heme species, we used the Rehm-Weller cycle. This analysis requires four energetic terms: 1) energy required to form radical heme species (based on iron-porphyrin systems⁵⁰) (1.7 V), 2) the ground state redox potential of OmcS (-212 mV)⁴⁹, 3) the photon energy used to excite OmcS nanowires ($\lambda = 545 \text{ nm} = 2.3 \text{ eV}$), and 4) the vibrational energy difference between the ground and excited states, called the Coulomb stabilization energy associated with the intermediate radical ion pair,⁵¹ (ω_p) $\sim 60 \text{ meV}$. Therefore, the energetics of this process would be $\Delta G_{\text{et}} = [1.7 \text{ eV} - (-0.212 \text{ eV}) + 0.06 \text{ eV}] - 2.3 \text{ eV} = -0.4 \text{ eV}$. Thus, $\Delta G_{\text{et}} < 0$ for the formation of the radical species. Our analysis is a lower estimate for the net energy available for the formation of the radical species. Therefore, in combination with our simulated analysis, our studies suggest that doubly oxidized species are Fe^{3+} + porphyrin radical and nanowires are photoreduced by ultrafast light-induced heme-to-heme charge transfer.

The authors also suggest a role for coherent delocalization of charge across multiple hemes (ref 50), which seems unlikely given the well understood and weak inter-heme couplings in these proteins, and the structural disorder that is present.

Our experiments are focused on excited state electron transfer whereas this reference was about ground-state electron transfer. Therefore, this reference has been removed to avoid confusion. We have also provided explicit analysis that rules out long-range coherence in OmcS (SI Fig. 9).

As well, the authors' claims seem to be inconsistent with finding of Blumberger and others for these multi-heme proteins. Subpicosecond heme-to-heme electron transfer claimed in the manuscript is at odds with recent studies of van Wonderen et al. (PNAS 118, e2107939118) that find nanosecond time scales for activationless heme-to-heme electron transfer.

We had discussed studies related to the paper mentioned by the reviewer in the introduction of the earlier version (Ref. 34-35 in the previous version).

We understand that the reviewers find our data inconsistent with sub-picosecond ground-state rates observed in these *monomeric* heme systems. However, we feel that this comparison is not appropriate because: (1) We report excited-state rates that are known to be faster due to the higher energy and larger orbital delocalization (Ref. 53) compared to ground-state rates measured in the above studies. Excited state charge transfers on a sub-picosecond scale have been reported extensively. For example, charge transfer between a donor-acceptor pair was previously reported on a 200-fs time scale (*JPCA* 106, 878, 2002). Charge transfer into hemes has been reported on a picosecond time scale (*PNAS* 112, 5602, 2015) and charge transfer in cryptochrome on a 400-fs time scale (Ref. 53). We have previously studied dye-sensitized large-bandgap semiconductors. In these systems, the charge must move from the dye, through the linker, into the conduction band of the semiconductor. These linkers are about 1 nm in length, significantly longer than the $\sim 3.5 \text{ \AA}$ heme-to-heme distance within OmcS and the orbitals have a less favorable energy alignment than two hemes have. Nevertheless, such dye-linker configurations still allowed a charge transfer on the sub-ps time scale (*JPCC* 120, 5940, 2016; *JPCC* 124, 3482, 2020).

(2) Furthermore, the cytochromes OmcS and MtrC are structurally different. For example, the slip-stack heme-to-heme distance in OmcS is smaller (3.5 Å) than the closest MtrC hemes 8/9 (4.3 Å). Computations have yielded an electron flux more than an order of magnitude larger in OmcS than through the Mtr system (JPCL11, 9421, 2020). We use the only heme system, with known structure, to be capable of polymerizing over micrometers with heme stacking closer than any previous systems. For closest-stacked hemes, our measured average ground-state rate of $3.2 \times 10^{10} \text{ s}^{-1}$ or a transfer time of $\sim 30 \text{ ps}$ (Ref. 52) is indeed comparable to the 15-90 ps rates reported in these previous studies; the fastest charge transfer measured experimentally in the van Wonderen paper, heme 8 to 9, has a rate constant of $11100 \times 10^6 \text{ s}^{-1}$ (Table 3 in Ref. 34) or a time of 90 ps. Blumberger has computed ground-state rate of $68,000 \times 10^6 \text{ s}^{-1}$ (Table 3 in Ref. 34) or a timescale of 15 ps.

Our ground-state rates are thus consistent with study mentioned by the reviewer whereas sub-picosecond rates in excited states could be key to the observed photoconductivity. We have further discussed this work as follows:

Furthermore, our studies show that sub-ps charge transfer is possible in natural proteins in an excited state. Prior ultrafast electron transfer studies have reported the fastest ground state rates of 15-90 ps in the closest-stacked hemes⁵⁸. This difference is likely because excited-state rates are known to be faster due to higher energy and larger orbital delocalization compared to ground-state rates⁵⁹.

My suspicion is that the photochemistry observed may involve multi-photon processes or redox active impurities in the samples, rather than direct heme-to-heme photoinduced charge transfer with unusual redox states (i.e., Fe(+4)).

Our power-dependent TA-measurements rule out multiphoton processes (Supplementary Figure 8). The electron transfer dynamics is independent of the laser power and the photon-flux, hence ruling out any significant contribution from multi-photon processes. The magnitude of photocurrent is also linear with increased power (Supplementary Figure 3).

Redox impurities do not contribute because of identical dynamics in solution and in solid-state (Supplementary Figure 7). Thus, the measured photophysical properties are intrinsic to the OmcS nanowires and not related to impurities. Furthermore, any redox active impurity would also need to be close to the hemes to affect electron transfer. However, the hemes in OmcS are shielded by the surrounding protein layer.

Furthermore, excitation of key amino acids such as tryptophan or tyrosine is not possible at the wavelengths used in this study and their molar extinction coefficient is 100-times smaller than the OmcS hemes. In the revised manuscript, we have discussed such possibilities as follows:

As no external electron donor was added, our results suggest that the additional electrons that reduce the heme are intrinsic to the nanowire itself. We further analyzed the possibility of protein surrounding to heme causing the observed photoreduction of OmcS hemes. Several aromatic amino acids, including tryptophan and tyrosine, are within 5 Å of the hemes in the OmcS. Although excitation of either tryptophan or tyrosine is not possible at the wavelengths used in this study^{44,45}, we considered a possibility that electron transfer can quench a photoexcited heme in a manner similar to flavins in a cryptochrome⁴⁸. This quenching would reduce a heme and leave behind an amino acid radical. The most likely amino acid candidate for radical formation is tryptophan because its radicals have absorbance in the wavelength range used in our experiments⁴⁹. While the formation of such radicals is possible, the signal strength in fs-TA measurements is determined by the molar extinction coefficients (ϵ) of the (transient) species. The molar extinction coefficient of OmcS's Soret band is approximately 100 times larger than those of tryptophan radicals^{49,50}. The ground state bleach represents all the photoexcited hemes in the OmcS nanowires and the species corresponding to $\lambda=367$ and 424 nm have differential absorption of

~20% and 10% of the total magnitude, respectively. Therefore, the number of tryptophan radicals created from electron transfer needs to be larger than the number of excited hemes in OmcS if the radical species at $\lambda=367$ nm arises from tryptophan. Such a possibility seems unlikely because only one radical can be created for every quenched excited heme. Thus, the observed spectra cannot be accounted for by amino acid radicals.

We also evaluated the possibilities of other electron sources causing photoreduction. We found that multi-photon processes are absent in our experiments because the ET dynamics were independent of the laser intensity and power (Supplementary Figure 8). The magnitude of photocurrent is also linear with increased power (Supplementary Figure 3).

Redox impurities also did not contribute to the measured spectra because of identical dynamics in solution and in solid-state (Supplementary Figure 7). Photodegradation also did not change the electron transfer dynamics, only the magnitude of spectra by $< 10\%$ over two hours. We therefore considered an alternative possibility that parallel-stacked hemes can serve as an electron donor and acceptor pair

Perhaps the processes at play arise from the photodegradation pathways cited in the manuscript.

Our experiments rule out such possibility. We would like to emphasize that the transient absorption is measured as a differential technique; meaning the difference of the excited minus the non-excited sample is measured 500 times per second (repetition rate of the laser is 1 kHz, every odd pulse is blocked). The measurements took at least one hour, the publication quality data in the main part were collected over more than 2.5 hours. This means that a permanent damage pathway cannot give a differential signal because the excited measurement is referenced onto the non-excited measurement collected one millisecond before. Damage can be observed when comparing the magnitude of the data measured after hours with the first measurement. We did observe a small decrease in signal (less than 10 % in 2 hours) which suggests moderate photodamage. However, photodamage only lowered the magnitude of the signal without changing the dynamics. As such photodegradation cannot account for the short time scales. The difference technique used in our study is solely sensitive to processes that are fully reversed within half a millisecond.

The events being studied, while interesting, may not originate in simple heme-to-heme charge flow at all.

The observed, up to 100-fold, increase in conductivity upon excitation cannot be explained by the other possibilities mentioned by the reviewer as discussed below:

(1) multi-photon processes absent are because the ET dynamics are independent of the laser intensity, and we observe power-independent ET dynamics. (2) Redox impurities do not contribute because we observe identical dynamics in solution and in solid-state and the signal from impurities is too small to account for observed changes. (3) Photodegradation did not change the electron transfer dynamics, only the magnitude of spectra by $< 10\%$ over two hours. (4) Excitation of either tryptophan or tyrosine is not possible at the wavelengths used in this study and their molar extinction coefficient is 100-times smaller than the OmcS hemes.

Typical nonradiative decay rates of heme singlet excited state are sub-nanosecond, so simple inter-heme electron electron transfer from heme excited states seems unlikely.

Sub-nanosecond rates cannot explain the observed sub-picosecond rates measured experimentally in OmcS. Furthermore, such rates will not give rise to the observed photoconductivity which requires charge separated states forming at a rate faster than the picosecond lifetime of the heme iron (Ref. 4).

Reviewer #2 (Remarks to the Author):

Referee report on revised manuscript “Microbial biofilms as living photoconductors due to ultrafast femtosecond heme-to-heme electron transfer in cytochrome nanowires” by Jens Neu et al., submitted to Nature Communications.

The initial title of the manuscript was “Living photoconductors based on ultrafast electron transfer in microbial nanowires”. From my perspective, the eye-catching highlight of this initial manuscript was “that the authors demonstrated photoconductivity in living biofilms of *Geobacter sulfurreducens*, which is not only a novelty from a fundamental point of view, but in the long term could also open the door to optoelectronic applications.

The claimed photoconductivity and the ultrafast electron transfer is obtained from a variety of state-of-the-art measurement techniques ranging from macroscopic interdigitated test structures to photoconductive atomic force microscopy and femtosecond transient absorption spectroscopy. “

Although an extensive – and still rapidly growing - amount of reports can be found on the electrical transport properties and electrical transport mechanisms in *Geobacter Sulfurreducens* and in other electro-active microorganisms, the study of the photoconductive properties is only starting – see next paragraph - and therefore highly novel.

Only a limited number of papers on the subject are found in literature, including the following : For *Geobacter Sulfurreducens*, Zhang et al. (ref. 20) demonstrated in 2021 that visible-light illumination could alter the electronic state of OM c-cyts from the ground state to the excited state in vivo. Van Wonderen, et al. (2019 – ref. 34) studied Ultrafast Light-Driven Electron Transfer in a Ru(II)tris(bipyridine)-^[SEP]Labeled Multiheme Cytochrome using^[SEP] ultrafast transient absorbance spectroscopy, to define heme-heme electron transfer dynamics in the representative multiheme cytochrome STC (tetra-heme Cyt_c) from *Shewanella oneidensis* in aqueous solution. STC was photosensitized by site-selective labeling with a Ru(II)(bipyridine)dye and the dynamics of light-driven electron transfer described by a kinetic model corroborated by molecular dynamics simulation and density functional theory calculations.

Furthermore, in 2021 Van Wonderen et al. reported transient absorbance spectroscopy when a His/Met-ligated heme was introduced at a defined site within the decaheme extracellular MtrC protein of *Shewanella Oneidensis*, observing rates of heme-to-heme electron transfer on the order of $10^{*}9s^{-1}$ (in good agreement with their predictions based on density functional and molecular dynamics calculations). This latter paper is not mentioned in the current manuscript of Jens Neu et al. and we'll come back to this paper in the next paragraphs.

We apologize for this oversight. In the earlier version, we had discussed the 2019 paper by Van Wonderen as well as comment on 2021 paper by Van Wonderen et al. mentioned by the reviewer Ref. 34-35). We have included discussion of this article in revised version (Ref. 34).

The current manuscript of Jens Neu et al., uses transient absorption spectroscopy (as van Wonderen et al.), but furthermore introduces photoconductive AFM (pc-AFM) in the study of photoconductivity of *Geobacter Sulfurreducens*. To the best of my knowledge, this is the very first report using pc-AFM on electro-active microorganisms. This innovative

combination of pc-AFM and the other mentioned techniques to explore the still unknown area of photoconductivity in *Geobacter Sulfurireducens*, as reported in the first part of the manuscript, is from my perspective the most appreciated part of the paper due to its high pioneering research level.

In the second part of the paper, the authors introduce TD-DFT calculations, quantum dynamics simulations, and propose a model for the origin of photoconductivity in protein nanowires, suggesting that upon photoexcitation, excited-state electron transfer is reducing the hemes in the nanowires with a “ultrafast femtosecond heme-to-heme electron transfer” as extra emphasized in the new title of the paper. (This “ultrafast femtosecond heme-to-heme electron transfer” in the new title is somewhat misleading as the used instrument response time is 100 +/- 15 fs.)

We apologize for this error. As per the reviewer’s suggestion, we have revised the manuscript title to sub-picosecond rather than femtosecond.

Furthermore, as also indicated by the other referee, this time-scale differs significantly from the “Nanosecond heme-to-heme electron transfer rates in multiheme cytochrome nanowire..” reported by van Wonderen et al. (PNAS2021-Vol.118-N°39-e2107939118), already mentioned above. An important drawback of the current version of the manuscript in relation to the proposed interpretation of the origin of the observed photoconductivity, is that a thorough discussion with these earlier results is missing, and that no reference is made to this paper of van Wonderen.

We apologize for this oversight. In the earlier version, we had discussed the 2019 paper by Van Wonderen as well as comment on 2021 paper by Van Wonderen et al. mentioned by the reviewer Ref. 34-35). We have included discussion of this article in revised version (Ref. 34).

As discussed in the response to the Reviewer-1, we understand that the reviewers find our data inconsistent with sub-picosecond ground-state rates observed in these *monomeric* heme systems. However, we feel that this comparison is not appropriate because:

(1) We report excited-state rates that are known to be faster due to the higher energy and larger orbital delocalization (Ref. 53) compared to ground-state rates measured in the above studies. Excited state charge transfers on a sub-picosecond scale have been reported extensively. For example, charge transfer between a donor-acceptor pair was previously reported on a 200-fs time scale (*JPCA* 106, 878, 2002). Charge transfer into hemes has been reported on a picosecond time scale (*PNAS* 112, 5602, 2015) and charge transfer in cryptochrome on a 400-fs time scale (Ref. 53). We have previously studied dye-sensitized large-bandgap semiconductors. In these systems, the charge must move from the dye, through the linker, into the conduction band of the semiconductor. These linkers are about 1 nm in length, significantly longer than the ~3.5 Å heme-to-heme distance within OmcS and the orbitals have a less favorable energy alignment than two hemes have. Nevertheless, such dye-linker configurations still allowed a charge transfer on the sub-ps time scale (*JPCC* 120, 5940, 2016; *JPCC* 124, 3482, 2020).

(2) Furthermore, the cytochromes OmcS and MtrC are structurally different. For example, the slip-stack heme-to-heme distance in OmcS is smaller (3.5 Å) than the closest MtrC hemes 8/9 (4.3 Å). Computations have yielded an electron flux more than an order of magnitude larger in OmcS than through the Mtr system (*JPCL* 11, 9421, 2020). OmcS is the only heme system, with an available structure, to be capable of polymerizing over micrometers. It shows heme stacking

closer than any previous systems. For the closest-stacked hemes, our measured average ground-state rate of $3.2 \times 10^{10} \text{ s}^{-1}$ or a transfer time of $\sim 30 \text{ ps}$ (Ref. 52) is indeed comparable to the 15-90 ps rates reported in these previous studies; the fastest charge transfer measured experimentally in the van Wonderen paper, heme 8 to 9, has a rate constant of $11100 \times 10^6 \text{ s}^{-1}$ (Table 3 in Ref. 34) or a time of 90 ps, and Blumberger and others have computed a rate of $68,000 \times 10^6 \text{ s}^{-1}$ (Table 3 in Ref. 34) or a time of 15 ps in the ground state.

We also agree with the other referee that to invoke cytochrome P450 chemistry (ref 46-47) to justify the generation of high oxidation state species (Fe(+4)) in the cytochromes under study in this paper, is doubtful due to structural and functional differences, and therefore possibly other processes are in play.

We apologize for this confusion. Although we did not directly claim that double oxidized species are Fe^{4+} , we recognize the confusion created by our citations that also show similarity to our spectra. We have removed these citations to avoid confusion. As discussed in the response to the Reviewer-1, We have performed additional analysis to identify the nature of the doubly oxidized species. Our analysis shows that the change in the spin populations occurs only on the ligands and not in the iron center. Therefore, our analysis suggests that doubly oxidized species are Fe^{3+} + porphyrin radical which agrees with the observed spectra at 367 nm. These analyses further suggest that the doubly oxidized species are **not Fe^{4+}** due to lack of change of spin density on the iron center upon additional oxidation of the ground state heme (Supplemental Figure 10). In addition, our energetic analysis using the Rehm-Weller cycle illustrates the thermodynamic feasibility of radical heme species. We have clarified these points in the revised manuscript.

The authors also state in the conclusions that “the surprising origin of the photoconductivity in these natural systems lies in the higher carrier density and mobility upon photoexcitation”. However the provided argumentation for an increased mobility is not clearly elaborated.

We have revised the Fig. 4 to highlight the likely cause for increased mobility. We have also clarified the discussion about increased mobility in the revised version as follows.

In addition to the higher carrier density due to photogenerated electrons, it is likely that the mobility of electrons increases upon photoexcitation due to increased driving force for charge transfer in the excited state of hemes⁶. Upon photoexcitation an electron is promoted from the ground state to an excited state. The ultrafast charge transfer between neighboring hemes creates a reduced-state heme in the excited state and a doubly oxidized heme (Fig. 4c-d). The reduced-state heme can then relax from the excited to the ground state. Upon photoexcitation, the uniformly oxidized nanowire is thus partially reduced and partially double oxidized (Fig. 4c).

The generated doubly oxidized heme will alter the redox energies of the heme chain, with more positive redox potential. We have previously found that the redox potential of OmcS hemes becomes substantially positive upon oxidation. The OmcS nanowires transport charges via a hopping mechanism⁷ - a process in which a charge (electron or hole) temporarily resides at a heme, changing its redox state. The driving force for charge transfer depends on the redox energies of the electron donating and accepting hemes. Therefore, the charge transfer rate is directly related to the mobility.

For the fully oxidized (non-excited) state, this process initiates at the electrode surface where injected electrons hop to nanowire redox sites, creating locally reduced hemes. For the photoexcited state, this process is enhanced because transferring an electron to the double-oxidized species, and removing an electron from a reduced heme, are significantly more favorable in the illuminated nanowire than for the oxidized nanowire in the dark. The increased likelihood for charge transfer upon photoexcitation will then result in increased mobility. Furthermore, the initial ultrafast charge transfer between hemes increases the

lifetime of the photogenerated state. Both the generation of a “new” mobile charge and the increase in its mobility will contribute to the observed increase in conductivity upon photoexcitation.

Also the hypothesis that “such ultrafast electron transfer might explain the quantum effects recently predicted in these OmcS nanowires (ref.50)” lacks argumentation.

Our experiments are focused on excited state electron transfer whereas this reference was about ground-state electron transfer. Therefore, this reference has been removed to avoid confusion.

We have also provided explicit analysis that rules out long-range coherence in OmcS (SI Fig. 9).

Although the first part of the paper was highly appreciated for its innovative experimental approach, the second part of the revised paper – although ambitious and with good intentions - does not provide at the moment sufficient convincing and strong arguments to support the formulated conclusions regarding the heme-heme electron transfer.

In conclusion, I would expect that a (major or new) revised version of the manuscript addressing the objections formulated above, could lead to a version suitable for publication in Nature Communications.

As suggested, we have performed several additional experiments and analysis and have thoroughly revised the manuscript to address all the concerns raised by the reviewers.

REVIEWERS' COMMENTS

Reviewer #2 (Remarks to the Author):

Referee report on “Microbial biofilms as living photoconductors due to ultrafast sub-picosecond electron transfer in cytochrome OmcS nanowires” by Jens Neu et al., submitted to Nature Communications

This is my 3rd referee report on this work. Also in my previous referee reports, I was in general very positive on the experimental outcome of this work: From my perspective, the eye-catching highlight of this work is “that the authors demonstrated photoconductivity in living biofilms of *Geobacter sulfurreducens*, which is not only a novelty from a fundamental point of view, but in the long term could also open the door to optoelectronic applications. The claimed photoconductivity and the ultrafast electron transfer is obtained from a variety of state-of-the-art measurement techniques ranging from macroscopic interdigitated test structures to photoconductive atomic force microscopy and femtosecond transient absorption spectroscopy. “ Although an extensive – and still rapidly growing - amount of reports can be found on the electrical transport properties and electrical transport mechanisms in *geobacter sulfurreducens* and in other electro-active microorganisms, the study of the photoconductive properties is only starting and therefore highly novel.

However, the earlier key concern - also emphasized by the other referee -, was the correct interpretation of the observed photoconductivity. In this version (3rd) of the manuscript the authors have made clear efforts to address the various concerns of the referees, including the ones considered as the most problematic ones by the other referee (eg. P450 chemistry and role of Fe⁴⁺, coherence,..). Significant changes have been made to the manuscript ; in particular the part with the discussion about “potential mechanisms” for the observed photoconductivity has been rewritten. For instance, additional analysis has been performed by the authors that suggests that the high oxidation species are not Fe⁴⁺, rather Fe³⁺. Also the earlier aspect of coherent delocalization – which was questioned by the referees – has been removed. In this version of the manuscript, the authors have addressed the main objections of the previous referee reports and provide an interpretation/hypothesis for the observed photoconductivity that with the current status of know how is plausible. This of course does not exclude that future insights or observations will yield alternative interpretations.

Minor remark : p. 7 the text of the figure caption (fig. 4d) is not complete : “..forming a doubly oxidized”

Based on all the arguments formulated in the report I am in favour to accept this revised version of the manuscript for publication in Nature Communications.

RESPONSE TO REVIEWERS' COMMENTS. Comments are in bold.

Reviewer #2 (Remarks to the Author):

Referee report on “Microbial biofilms as living photoconductors due to ultrafast sub-picosecond electron transfer in cytochrome OmcS nanowires” by Jens Neu et al., submitted to Nature Communications

This is my 3rd referee report on this work. Also in my previous referee reports, I was in general very positive on the experimental outcome of this work: From my perspective, the eyecatching highlight of this work is “that the authors demonstrated photoconductivity in living biofilms of Geobacter sulfurreducens, which is not only a novelty from a fundamental point of view, but in the long term could also open the door to optoelectronic applications. The claimed photoconductivity and the ultrafast electron transfer is obtained from a variety of state-of-the-art measurement techniques ranging from macroscopic interdigitated test structures to photoconductive atomic force microscopy and femtosecond transient absorption spectroscopy. “ Although an extensive – and still rapidly growing - amount of reports can be found on the electrical transport properties and electrical transport mechanisms in geobacter sulfurreducens and in other electro-active microorganisms, the study of the photoconductive properties in only starting and therefore highly novel.

However, the earlier key concern - also emphasized by the other referee -, was the correct interpretation of the observed photoconductivity. In this version (3rd) of the manuscript the authors have made clear efforts to address the various concerns of the referees, including the ones considered as the most problematic ones by the other referee (eg. P450 chemistry and role of Fe⁴⁺, coherence,..). Significant changes have been made to the manuscript ; in particular the part with the discussion about ”potential mechanisms” for the observed photoconductivity has been rewritten. For instance, additional analysis has been performed by the authors that suggests that the high oxidation species are not Fe⁴⁺, rather Fe³⁺. Also the earlier aspect of coherent delocalization – which was questioned by the referees – has been removed. In this version of the manuscript, the authors have addressed the main objections of the previous referee reports and provide an interpretation/hypothesis for the observed photoconductivity that with the current status of know how is plausible. This of course does not exclude that future insights or observations will yield alternative interpretations.

Minor remark : p. 7 the text of the figure caption (fig. 4d) is not complete : “..forming a doubly oxidized”

We have ensured that the text is complete for all figure captions.

Based on all the arguments formulated in the report I am in favour to accept this revised version of the manuscript for publication in Nature Communications.